# Trace Reconstruction with Language Models

**Franziska Weindel**[*]                   *franziska.weindel@tum.de*
*Technical University of Munich*
*Munich Center for Machine Learning*

**Michael Girsch**[*]                    *michael.girsch@tum.de*
*Technical University of Munich*
*Munich Center for Machine Learning*

**Reinhard Heckel**[†]                   *reinhard.heckel@tum.de*
*Technical University of Munich*
*Munich Center for Machine Learning*

**Reviewed on OpenReview:** *https://openreview.net/forum?id=k2zAyJAfnj*

## Abstract

The general trace reconstruction problem seeks to recover an original sequence from its noisy copies independently corrupted by insertions, deletions, and substitutions. This problem arises in applications such as DNA data storage, a promising storage medium due to its high information density and longevity. However, errors introduced during DNA synthesis, storage, and sequencing require correction through algorithms and codes, with trace reconstruction often used as part of data retrieval. In this work, we propose TReconLM, a decoder-only transformer that solves trace reconstruction as a next-token prediction task. TReconLM outperforms state-of-the-art trace reconstruction algorithms, including prior deep-learning approaches, recovering a substantially higher fraction of sequences without error. We pretrain on synthetic data generated from a simple error model and fine-tune on real-world data to adapt to technology-specific error patterns. Code is available at `https://github.com/MLI-lab/TReconLM`.

## 1 Introduction

Trace reconstruction is a central problem in biological data analysis (Antkowiak et al., 2020; Bar-Lev et al., 2025; Organick et al., 2018). Given multiple noisy copies of a sequence (traces), the goal is to reconstruct the original sequence from as few traces as possible.

For example, in DNA data storage, the sequences to be reconstructed typically consist of 50-200 nucleotides of adenine (A), cytosine (C), guanine (G), and thymine (T). For some sequences, as few as 2-10 noisy traces are available, each independently corrupted by insertions, deletions, and substitutions. However, existing trace reconstruction methods, including general algorithms such as MUSCLE (Edgar, 2004) as well as algorithms specifically developed for DNA data storage (Qin et al., 2024), struggle when few traces are available and error rates are high.

Existing trace reconstruction algorithms either assume a fixed error model (Srinivasavaradhan et al., 2021; Viswanathan & Swaminathan, 2008) or rely on observed traces, often using dynamic programming techniques such as computing the longest common subsequence (Edgar, 2004; Gopalan et al., 2018; Sabary et al., 2024). Fixed error models fail to capture error dependencies observed in practice, such as error probabilities increasing with sequence length (Gimpel et al., 2023). Relying only on observed traces ignores known

---

[*]Equal contribution
[†]Correspondence to: `reinhard.heckel@tum.de`

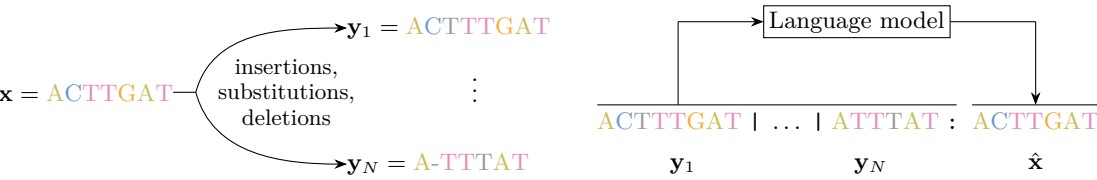

Figure 1: Left: Trace reconstruction aims to recover a sequence $x$ from $N$ noisy copies $y_i$, each corrupted by insertions, deletions, and substitutions. Right: We formulate trace reconstruction as a next-token prediction task and train a transformer model to reconstruct $x$ from its noisy traces.

error statistics, which can provide useful prior information, especially when few traces are available. These limitations motivate a data-driven approach that can be trained on synthetic data generated from an error model and fine-tuned on real-world data to capture observed error dependencies.

In this work, we formulate the trace reconstruction problem for DNA data storage as a next-token prediction task and train a decoder-only transformer to generate sequence estimates from a set of traces. Our method, TReconLM (Trace Reconstruction with a Language Model), outperforms existing state-of-the-art approaches for reconstructing DNA sequences from few traces, including both classical methods and specialized deep-learning architectures. To address the lack of large-scale real data, we pretrain TReconLM on synthetic data generated with a technology-agnostic error model and fine-tune on data from existing DNA data storage systems to mitigate distribution shifts and improve performance.

In addition to providing a simple, state-of-the-art method, we analyze why and how transformers learn to perform trace reconstruction so well. First, we study scaling laws to understand how model size affects performance. We find that relatively small models (e.g., 38M parameters) perform best, and that increasing the model size further does not improve performance. We support this empirical finding with a theoretical analysis that explains the behavior. Second, we show that transformers can express the Bayes-optimal algorithm under substitution errors and that any transformer achieving near-optimal loss must approximate it.

Perhaps surprisingly, framing trace reconstruction as a next-token prediction task and training a standard decoder-only transformer outperforms existing approaches on this challenging algorithmic problem. This highlights the potential of language models for algorithmic tasks and contributes to emerging literature showing that signal reconstruction problems can be effectively solved with learning-based approaches.

## 2 Related work

Theoretical work on the trace reconstruction problem typically studies the minimum number of traces required to reconstruct a binary sequence corrupted by deletions, with high probability (Chase, 2021; De et al., 2017; Holden & Lyons, 2020). However, perfect reconstruction from a small number of traces, which is the regime we focus on in this paper, is generally not possible.

Several trace reconstruction methods have been proposed in previous work. For traces corrupted by deletions, Batu et al. (2004) introduced the bitwise majority alignment (BMA) algorithm, which uses symbol-wise majority voting. Viswanathan & Swaminathan (2008) extended BMA for insertions, deletions, and substitutions, and Gopalan et al. (2018) proposed another BMA-based method.

Antkowiak et al. (2020) performed trace reconstruction using multiple sequence alignment (MSA) with the MUSCLE algorithm (Edgar, 2004), followed by majority voting across alignment columns. Sabary et al. (2024) proposed several dynamic programming-based methods, including shortest common supersequence and longest common subsequence algorithms. Their iterative algorithm (ITR) achieves state-of-the-art performance for trace reconstruction in DNA data storage. Srinivasavaradhan et al. (2021) introduced Trellis-BMA, which combines the BCJR algorithm (Bahl et al., 1974) with BMA-based methods.

Qin et al. (2024) proposed RobuSeqNet, a neural network-based approach that combines an attention mechanism, a conformer encoder, and an LSTM decoder. Input sequences are one-hot encoded, padded to a fixed length, and represented as matrices, which are then aggregated across traces. The attention module downweights misclustered sequences. On large clusters, RobuSeqNet achieves similar performance to the state-of-the-art ITR algorithm.

Bar-Lev et al. (2025) proposed DNAformer, an end-to-end DNA data storage framework that includes a coding scheme and a transformer-based trace reconstruction model. Their neural architecture differs from ours in several key aspects. First, as in Qin et al. (2024), input sequences are one-hot encoded. Second, the model uses a two-branch structure with shared weights processing forward and reversed sequences. Third, DNAformer includes a learned alignment module, followed by a transformer encoder without positional embeddings or causal masks, and uses dynamic programming as an optional postprocessing step. DNAformer achieves similar performance to the ITR algorithm.

Nahum et al. (2021) proposed a sequence-to-sequence transformer for single-read reconstruction, where noisy sequences are grouped by length and processed by separate transformer networks. Their model acts as a sequence classifier that maps each noisy sequence to one of 256 predefined codewords. In contrast, our work uses next-token prediction instead of sequence-to-sequence mapping.

Dotan et al. (2023) introduced BetaAlign, an encoder-decoder transformer for aligning biological sequences.

## 3 Background and problem statement

In this paper, we study the trace reconstruction problem. Given a set of noisy copies (traces) $\boldsymbol{y}_1, \ldots, \boldsymbol{y}_N$ of a sequence $\boldsymbol{x}$, independently corrupted by unknown insertions, deletions, and substitutions, the goal is to compute a sequence estimate $\hat{\boldsymbol{x}}$ of the original sequence $\boldsymbol{x}$. Figure 1, left panel, illustrates the problem statement.

We focus on the trace reconstruction problem in DNA data storage, where the sequence $\boldsymbol{x}$ is a DNA strand of length 50-200 nucleotides that can be modeled as a random sequence over the quaternary alphabet. To motivate this setting, we briefly outline the DNA data storage pipeline.

In DNA data storage, digital information is first partitioned into short segments to accommodate current synthesis limitations, which prevent reliably writing long DNA strands. Each segment is then encoded using an error-correcting code and mapped to a DNA sequence $\boldsymbol{x}_i \in \{\mathrm{A, C, G, T}\}^L$ of length $L$, resulting in a set $\mathcal{D} = \{\boldsymbol{x}_1, \ldots, \boldsymbol{x}_M\}$. The sequences in $\mathcal{D}$ are synthesized, amplified, and can be stored over long periods.

At readout, a subset of DNA sequences is sampled and sequenced, resulting in multiple unordered, noisy traces of the original sequences. The number of traces per sequence varies due to amplification bias and random sampling.

To recover the stored information, the first step is typically clustering, where the goal is to group traces originating from the same sequence $\boldsymbol{x}_i$. However, clustering is imperfect: a single original sequence can give rise to multiple clusters, and a single cluster can have traces from different original sequences (Antkowiak et al., 2020; Organick et al., 2018; Rashtchian et al., 2017).

After clustering, the next step is to reconstruct a candidate sequence for each cluster. Given a cluster containing $N \in \mathbb{N}_0$ noisy copies $\boldsymbol{y}_1, \ldots, \boldsymbol{y}_N$ of a DNA sequence $\boldsymbol{x}_i$, each independently corrupted by unknown insertions, deletions, and substitutions, the goal is to compute a sequence estimate $\hat{\boldsymbol{x}}_i$ of the original sequence $\boldsymbol{x}_i$. This is the trace reconstruction problem defined at the beginning of this section.

We assume that the sequences $\boldsymbol{x}_i$ consist of nucleotides chosen uniformly at random over the alphabet $\{\mathrm{A, C, G, T}\}$. This assumption is reasonable, as many DNA data storage systems add a pseudorandom sequence to the input data before encoding (Antkowiak et al., 2020; Organick et al., 2018), and compressed data is approximately uniformly distributed (Cover, 1999). After trace reconstruction, a decoder typically corrects any remaining errors in the sequence estimates $\hat{\boldsymbol{x}}_i$ using the redundancy from encoding to recover the original information.

## 4 Method

We formulate the trace reconstruction problem as a next-token prediction task and train a decoder-only transformer to solve it. Given a set of $N$ traces

$$\mathcal{C} = \{\boldsymbol{y}_1, \ldots, \boldsymbol{y}_N\},$$

we train a model $f_{\boldsymbol{\theta}}$ with parameters $\boldsymbol{\theta}$ to predict an estimate $\hat{\boldsymbol{x}}$ of the original sequence $\boldsymbol{x}$ of length $L$ when prompted with the concatenation of traces

$$\boldsymbol{p} = \boldsymbol{y}_1 \mid \boldsymbol{y}_2 \mid \ldots \mid \boldsymbol{y}_{N-1} \mid \boldsymbol{y}_N : . \tag{1}$$

We introduce the $\mid$ token to concatenate the traces and the $:$ token to mark the end of all traces. The model's vocabulary is $\mathcal{V} = \{\mathrm{A}, \mathrm{C}, \mathrm{G}, \mathrm{T}, \mid, :, \#\}$, where $\#$ is the padding token.

Given prompt $\boldsymbol{p}$ as in Equation 1, the model autoregressively generates $L$ tokens using greedy decoding to obtain the sequence estimate $\hat{\boldsymbol{x}}$. See Figure 1, right panel, for an illustration. In Appendices A and B, we compare against alignment-based targets and alternative decoding strategies, finding that direct prediction with greedy decoding performs best.

### 4.1 Training and data generation

We generate synthetic data by first sampling an original sequence $\boldsymbol{x} \in \{\mathrm{A}, \mathrm{C}, \mathrm{G}, \mathrm{T}\}^L$ of length $L$ uniformly at random. Each trace $\boldsymbol{y}_j$ is then generated by passing $\boldsymbol{x}$ through a noisy channel that processes each base sequentially with four possible outcomes: deletion (probability $p_{\mathrm{D}}$, delete base), insertion (probability $p_{\mathrm{I}}$, add a random base and reprocess the current base), substitution (probability $p_{\mathrm{S}}$, replace it with a different random base), or correct transmission (probability $p_{\mathrm{T}} = 1 - p_{\mathrm{D}} - p_{\mathrm{I}} - p_{\mathrm{S}}$, copy the base unchanged). While more complex error distributions are possible (e.g., position-dependent or context-dependent), we use a uniform error model for pretraining, since it is technology-agnostic and approximates real-world errors well (Gimpel et al., 2023). In Appendix J.2, we show that this approach is also effective, outperforming DNAformer (Bar-Lev et al., 2025) trained directly on real-world error statistics.

The traces $\boldsymbol{y}_1, \ldots, \boldsymbol{y}_N$ are concatenated with the original sequence to form a training instance

$$\boldsymbol{y}_1 \mid \boldsymbol{y}_2 \mid \ldots \mid \boldsymbol{y}_{N-1} \mid \boldsymbol{y}_N : \boldsymbol{x}. \tag{2}$$

For each training instance, we randomly permute the order of traces, sample the error probabilities uniformly from $[0.01, 0.1]$, and draw the number of traces $N$ uniformly from $[2, 10]$, as this represents a practically relevant and challenging regime.

The transformer model is trained on synthetic data by minimizing cross-entropy loss between the predicted sequence $\hat{\boldsymbol{x}}$ and the original sequence $\boldsymbol{x}$.

### 4.2 Fine-tuning on real data

In practice, error probabilities are often correlated and may vary with the position in the DNA sequence, leading to a distribution shift between our synthetic training data and real-world data. To mitigate this shift, we fine-tune on real-world datasets (Antkowiak et al., 2020; Srinivasavaradhan et al., 2021; Chandak et al., 2020), as discussed in Sections 5.3.1, 5.3.2, and Appendix I.

For each ground-truth sequence $\boldsymbol{x}$ in the datasets, we associate noisy traces $\boldsymbol{y}_1, \ldots, \boldsymbol{y}_N$ to construct training examples as in Equation 2. We then fine-tune the model analogously to pretraining. Fine-tuning on data from a single storage experiment results in a technology-adapted model that can be used for reconstruction in subsequent experiments with the same sequence length and read/write setup.

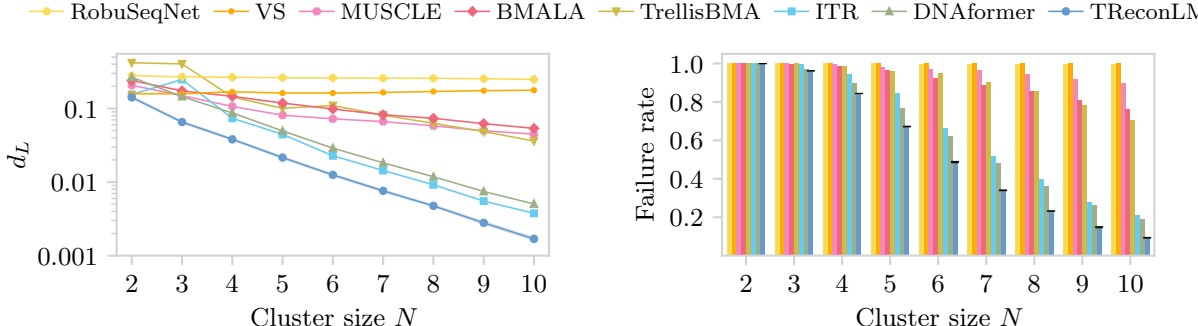

Figure 2: Average Levenshtein distances $d_L$ and failure rates on synthetic data with sequence length $L = 110$. TReconLM is averaged over three runs with different seeds. Shaded bands and error bars show $\pm$ one standard deviation, which is very small and hardly visible (on average, 0.11% for failure rate and $< 0.0001$ for Levenshtein distance).

## 5 Experiments

In this section, we evaluate our proposed method TReconLM on both synthetic and real-world data from existing DNA data storage systems. We find that TReconLM outperforms the state-of-the-art ITR algorithm (Sabary et al., 2024) across all evaluated regimes.

We measure performance using the following two metrics:

- **Levenshtein distance** $d_{\mathrm{L}}(\boldsymbol{x}, \hat{\boldsymbol{x}})$: The minimum number of edits (insertions, deletions, and substitutions) required to transform the reconstructed sequence $\hat{\boldsymbol{x}}$ into the original sequence $\boldsymbol{x}$, normalized by the length $L$ of the original sequence.

- **Failure rate**: The fraction of test examples in which the reconstructed sequence $\hat{\boldsymbol{x}}$ differs from the original sequence $\boldsymbol{x}$.

### 5.1 Baselines

We compare TReconLM to both dynamic programming-based and deep-learning-based reconstruction methods. For dynamic programming-based methods, we consider the ITR algorithm (Sabary et al., 2024), trace reconstruction using MUSCLE (Edgar, 2004) with majority voting, TrellisBMA (Srinivasavaradhan et al., 2021), BMALA (Gopalan et al., 2018), and VS (Viswanathan & Swaminathan, 2008). For deep-learning-based methods, we consider RobuSeqNet (Qin et al., 2024) and DNAformer (Bar-Lev et al., 2025). Descriptions of all baselines and implementation details are given in Appendix K. In Appendix J.3, we additionally compare TReconLM with GPT-4o mini and GPT-5 under zero- and few-shot prompting, with and without Chain-of-Thought reasoning.

### 5.2 Evaluation on synthetic data

We first evaluate reconstruction performance on synthetic data generated as described in Section 4.1 for three sequence lengths $L = 60$, 110, and 180, which are representative of existing DNA data storage systems.

We train a decoder-only transformer model with $\sim$38M parameters on $\sim$300M examples ($\sim$440B tokens), totaling $1.0 \times 10^{20}$ FLOPs. We choose the model size based on the scaling law analysis in Section 5.4, which shows that increasing the number of parameters further does not improve performance.

We train our deep-learning baselines, RobuSeqNet ($\sim$3M parameters) and DNAformer ($\sim$100M parameters), on the same training set. We do not match compute since RobuSeqNet's smaller model size would require significantly longer training. In Appendix J, we show that TReconLM also outperforms RobuSeqNet when

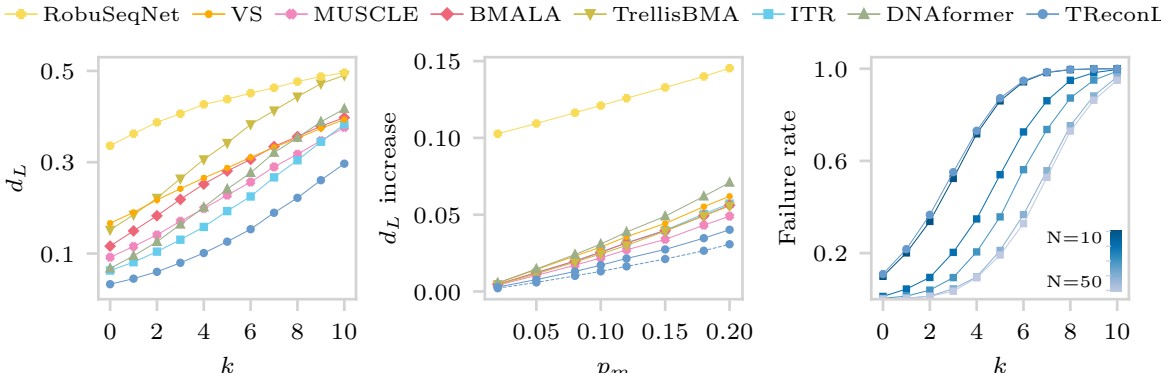

Figure 3: Left: Average Levenshtein distances $d_L$ under increasing noise levels $k$. Middle: Increase in average $d_L$ (relative to no misclustering) when each trace is replaced by a misclustered one with probability $p_m$. The dashed line shows TReconLM trained on misclustered data. Right: Failure rates of TReconLM under increasing noise levels when trained with fixed cluster size $N$.

controlling for compute and model size, and DNAformer based on the self-reported performance in Bar-Lev et al. (2025). We evaluate all algorithms on a shared test set of 50K randomly generated examples, constructed in the same way as the training set.

Figure 2 shows the average Levenshtein distances and failure rates. TReconLM outperforms all baseline methods across all cluster sizes $N \in [2, 10]$, reconstructing 11.9% more sequences than the state-of-the-art ITR algorithm. In Appendix C, we show that TReconLM also outperforms all baselines for shorter ($L = 60$) and longer ($L = 180$) sequences. In Appendix D, we analyze failure modes and find that consecutive repeated nucleotides have a 1.41 times higher error rate than expected under uniform difficulty across patterns, which we explain by the fact that they occur less frequently under our data generation.

### 5.2.1 Robustness to higher noise levels

We next evaluate the robustness of TReconLM to higher noise levels. We sweep over noise levels, sampling insertion, deletion, and substitution probabilities uniformly from $[0.01 + 0.01k, 0.10 + 0.01k]$ for $k = 0, \ldots, 10$.

Figure 3, left panel, shows the average Levenshtein distances between reconstructed and ground-truth sequences for TReconLM and baselines, evaluated on a shared test set of 5K randomly sampled examples per noise level $k$. TReconLM achieves lower Levenshtein distances across all noise levels, outperforming the baselines despite a mismatch between training and test error distributions.

### 5.2.2 Robustness to misclustered sequences

In practice, clustering algorithms are imperfect and may assign traces from different ground-truth sequences to the same cluster. We evaluate robustness to misclustering by corrupting the test set from Section 5.2. For each trace, we independently replace it with a misclustered sequence with probability $p_m$ ranging from 0.02 to 0.20. We generate misclustered sequences by sampling a random ground-truth and applying the same noise distribution as during training.

Figure 3, center panel, shows the increase in average Levenshtein distance compared to the test set without misclustering, for all methods and misclustering rates $p_m$. TReconLM shows the smallest increase compared to baselines, and training on misclustered data (dashed line) further improves robustness. For additional details, see Appendix E.

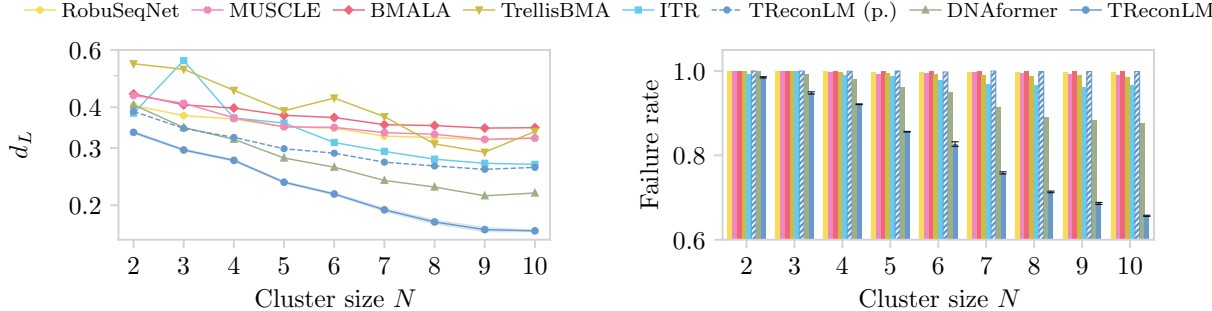

Figure 4: Average Levenshtein distances $d_L$ and failure rates on the out-of-distribution Noisy-DNA dataset. Fine-tuned TReconLM is the average of three runs with different seeds from the same pretrained model, with shaded bands and error bars showing $\pm$ one standard deviation.

### 5.2.3 Performance under larger and fixed cluster sizes

To understand how performance scales with cluster size, we train separate models with fixed cluster sizes $N \in \{10, 20, \ldots, 50\}$. Each model is trained on error probabilities sampled uniformly from $[0.01, 0.10]$ to reconstruct sequences of length $L = 110$ with a compute budget of $1.0 \times 10^{20}$ FLOPs. We evaluate the models on 5K test sequences at each noise level $k$.

Figure 3, right panel, shows that increasing the cluster size from $N = 10$ to $N = 20$ improves reconstruction performance, with diminishing returns for larger cluster sizes. The panel also compares our model from Section 5.2 (pretrained on varying cluster sizes $N \in [2, 10]$) with a model trained on a fixed cluster size $N = 10$; we find only a small performance decrease when training on varying cluster sizes.

### 5.2.4 Generalization across sequence lengths

To test whether TReconLM can generalize across sequence lengths, we train a model on varying sequence lengths sampled uniformly from $[50, 120]$, using the same compute budget of $1.0 \times 10^{20}$ FLOPs as the fixed-length models from Section 5.2. We evaluate it on sequences of length $L = 60$ and $L = 110$ (in-distribution) and $L = 180$ (out-of-distribution).

The variable-length model performs comparably to the fixed-length models on in-distribution lengths, with Levenshtein distance increases of 0.004 and 0.01 for $L = 60$ and $L = 110$, respectively, and failure rate increases of 0.012 and 0.089. We expect these gaps to close with more training data. However, performance at $L = 180$ is much worse (Levenshtein distance increase of 0.45 and failure rate increase of 0.38), primarily due to early sequence termination: the model generates 141 tokens on average versus the target of 180. When constrained to generate 180 tokens, the model defaults to repetitive patterns (e.g., `ATAT`). We attribute this to the positional embeddings for tokens beyond the training lengths never receiving gradient updates. These results suggest that the model generalizes well within the distribution of training lengths but not beyond. In practice, DNA data storage systems use fixed-length sequences, so a single model trained on the full range of sequence lengths used by current technologies (50-200 nucleotides) suffices.

### 5.3 Experiments on real data

In this section, we show that fine-tuning on real-world data improves reconstruction performance relative to pretrained models. We evaluate on datasets from DNA data storage systems using cost-efficient technologies with high error probabilities: the Noisy-DNA dataset (Antkowiak et al., 2020), which uses light-based synthesis, and the Microsoft dataset (Srinivasavaradhan et al., 2021), which uses nanopore sequencing. Most other existing DNA storage systems use higher-quality but more expensive technologies (Goldman et al., 2013; Erlich & Zielinski, 2017; Organick et al., 2018; Grass et al., 2015), where existing algorithms already perform well. In Appendix I, we additionally consider the dataset from Chandak et al. (2020), and in

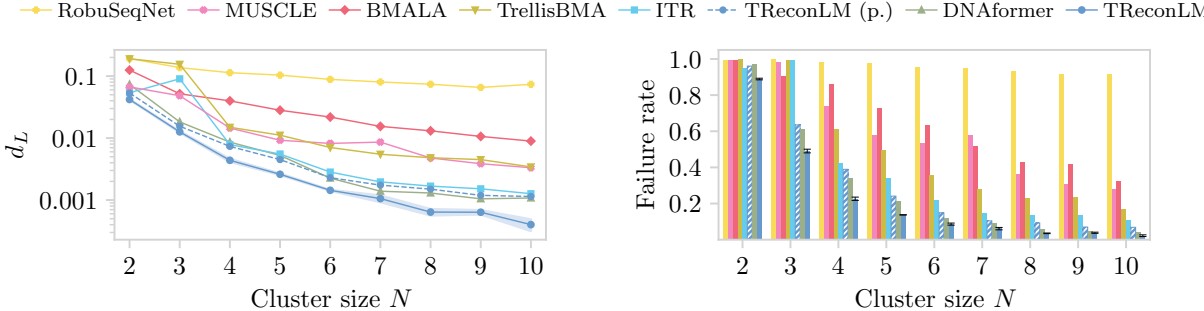

Figure 5: Average Levenshtein distances $d_L$ and failure rates on the out-of-distribution Microsoft dataset. Fine-tuned TReconLM is the average of three runs with different seeds from the same pretrained model, with shaded bands and error bars showing $\pm$ one standard deviation.

Appendices F and H, we show that pretraining improves performance over training directly on real-world data, and analyze fine-tuning data efficiency.

### 5.3.1 Real data experiment 1: Noisy-DNA dataset

The Noisy-DNA dataset (Antkowiak et al., 2020) consists of $M = 16{,}383$ ground-truth sequences of length $L = 60$ nucleotides and their unclustered traces. Estimated error probabilities are $p_I = 0.057$ (insertions), $p_D = 0.06$ (deletions), and $p_S = 0.026$ (substitutions), significantly higher than in other DNA data storage systems (Goldman et al., 2013; Grass et al., 2015; Erlich & Zielinski, 2017; Organick et al., 2018). The error rates are position-dependent, with the insertion probability increasing toward the end of the sequences to $p_I = 0.3$. This makes the dataset well-suited for evaluating whether fine-tuning can adapt to real-world error statistics.

We construct our fine-tuning dataset by clustering traces by sequence index and discarding traces with index errors. More advanced clustering methods (Zorita et al., 2015) could further reduce failure rates, but our goal is to compare the relative performance of reconstruction methods.

After clustering, we split the dataset into 80% train, 10% validation, and 10% test clusters. Clusters with more than ten traces are split into smaller subclusters to fit within the model's context window. For the validation and test sets, we precompute fixed subclusters with sizes between 2 and 10, giving 15,578 validation and 15,696 test examples. For training, we apply dynamic subclustering: in each epoch, we sample up to 10 traces per cluster in random order, so the model sees different subsets and permutations across epochs. To prevent data leakage, we remove traces that are too similar to test-set ground-truth sequences based on Levenshtein distance. Details of the preprocessing pipeline are provided in Appendix G.

We fine-tune the pretrained TReconLM for a compute budget of $1 \times 10^{18}$ FLOPs, and the pretrained DNAformer and RobuSeqNet for an equivalent number of training steps for a fair comparison. Figure 4 shows average Levenshtein distances and failure rates for different cluster sizes. While the pretrained model fails to generalize to technology-dependent error statistics, fine-tuning significantly improves performance. TReconLM recovers 14.7% more sequences than the state-of-the-art ITR algorithm, a larger improvement over ITR than on synthetic data (11.9%). This is consistent with its robustness to misclustering (Section 5.2.2), as the simple index-based approach can introduce clustering errors.

### 5.3.2 Real data experiment 2: Microsoft dataset

We next evaluate the performance of TReconLM on the Microsoft dataset from Srinivasavaradhan et al. (2021), which consists of $M = 10{,}000$ ground-truth sequences of length $L = 110$ and 269,707 traces. The traces are pre-clustered using the algorithm from Rashtchian et al. (2017), with each cluster corresponding to a single ground-truth sequence. Estimated error probabilities are $p_I = 0.017$ (insertions), $p_D = 0.02$ (deletions), and $p_S = 0.022$ (substitutions).

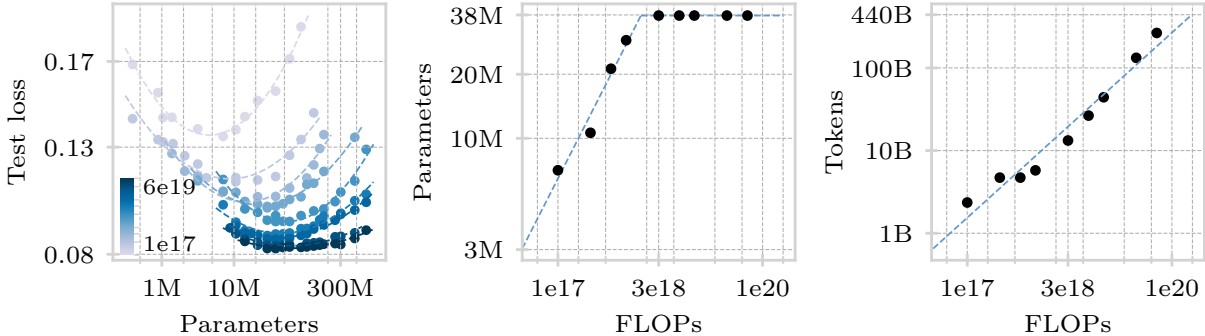

Figure 6: IsoFLOP curves for trace reconstruction. Left: Test loss for models ranging from 450K to 680M parameters, trained on sequences of length $L = 110$ across nine compute budgets from $10^{17}$ to $6 \times 10^{19}$ FLOPs. Center: Number of parameters of the best-performing models versus compute, showing a plateau at high compute budgets. Right: Number of tokens versus compute, following scaling laws observed in language modeling.

As with the Noisy-DNA dataset (Antkowiak et al., 2020), we split the data into 80% train, 10% validation, and 10% test clusters. Subclusters are precomputed for the validation and test sets and dynamically sampled during training to fit within the model's context length. We obtain 4,977 and 5,109 examples for the validation and test sets, respectively, each with cluster sizes $N \leq 10$. For the train set, we obtain 7,976 examples with cluster sizes $N \in \mathbb{N}_0$.

Figure 5 shows reconstruction performance after fine-tuning TReconLM from Section 5.2 using a compute budget of $1 \times 10^{19}$ FLOPs. The pretrained RobuSeqNet and DNAformer models are fine-tuned on the same dataset for an equivalent number of steps. Both the pretrained and fine-tuned TReconLM models outperform all non-deep-learning baselines. The pretrained TReconLM performs comparably to the fine-tuned DNAformer, suggesting that TReconLM can generalize to some extent from synthetic data to the Microsoft dataset, despite differences in error statistics.

### 5.4 Scaling laws for trace reconstruction

We study how performance scales with compute and determine the best model size for reconstructing sequences of length $L = 110$ at a fixed compute budget. Following Approach 2 of Hoffmann et al. (2022), we train a suite of models ranging from $N_\mathrm{p} = 450\mathrm{K}$ to 680M parameters at nine compute budgets $C \in \{1, 3, 6\} \times 10^{17,18,19}$ FLOPs, where each model is trained on $T = \frac{C}{6N_\mathrm{p}}$ tokens.

We fix all optimization hyperparameters across runs, varying only the batch size (between 8 and 1.2K) and scaling the learning rate accordingly. The embedding dimension to depth ratio ranges from approximately 28 to 122. Other optimization hyperparameters are listed in Table 1 in Appendix C.

Figure 6 shows the IsoFLOP curves across the considered compute budgets, plotting test loss against model size (log scale, left panel). Under our hyperparameter settings, we observe that after a compute budget of $3 \times 10^{18}$ FLOPs, the optimal model size plateaus at approximately 38M parameters (center panel). We provide a possible theoretical explanation for this behavior in Section 6. The number of tokens processed versus compute follows a standard power law relationship (right panel).

## 6 Theory

To understand the scaling behavior in Figure 6 and to probe whether TReconLM approximates the Bayes-optimal estimator, we analyze a simplified setting with only substitution errors. We focus on this setting because the optimal estimator has a simple closed form (majority vote), whereas for insertions and deletions no closed-form optimal estimator is known.

## 6.1 Scaling behavior under substitution-only errors

We consider a sequence $\tilde{\boldsymbol{x}} \in \{-1, 1\}^n$ and assume that we have $m$ noisy copies of the sequence $\tilde{\boldsymbol{x}}_1, \ldots, \tilde{\boldsymbol{x}}_m$, where each copy is obtained by independently flipping each of the entries in $\tilde{\boldsymbol{x}}$ with probability $p < 1/2$. Our goal is to estimate the sequence $\tilde{\boldsymbol{x}}$ based on the noisy copies. This is a special case of the trace reconstruction problem considered in this paper, where we only have substitutions, as opposed to substitutions, deletions, and insertions.

We consider a linear estimator with $kn \leq mn$ parameters for estimating each position of $\boldsymbol{x}$, with weights bounded by $\|\boldsymbol{w}\|_2 \leq B = \sqrt{kn}$ and $\|\boldsymbol{x}\|_2 \leq R = \sqrt{kn}$. Without loss of generality, we focus on estimating the first position of the sequence, $y = [\tilde{\boldsymbol{x}}]_1$. Our estimator takes the form

$$\hat{y} = \text{sign}\left(\sum_{i=1}^{k} \boldsymbol{w}_i^T \tilde{\boldsymbol{x}}_i\right) = \text{sign}(\boldsymbol{w}^T \boldsymbol{x}).$$

The Bayes-optimal estimator only uses the coordinates $[\tilde{\boldsymbol{x}}_1]_1, \ldots, [\tilde{\boldsymbol{x}}_k]_1$ since the other coordinates are independent of $[\tilde{\boldsymbol{x}}]_1$, and is $\boldsymbol{w}_B = [1, 0, ..., 1, 0, ..., 1, 0, ...]$ (or a scaled version thereof).

We perform logistic regression to learn an estimator of the form $\text{sign}(\boldsymbol{w}^T \boldsymbol{x})$ from $N$ examples

$$\hat{\boldsymbol{w}} = \arg\min_{\boldsymbol{w} \,:\, \|\boldsymbol{w}\|_2 \leq B} \hat{R}(\boldsymbol{w}), \quad \text{where} \quad \hat{R}(\boldsymbol{w}) = \frac{1}{N} \sum_{i=1}^{N} \ell(\boldsymbol{w}^T \boldsymbol{x}_i, y_i), \tag{3}$$

where $\ell(z, y) = \log(1 + e^{-yz})$ is the logistic loss.

**Proposition 1.** *With probability at least $1 - \delta$, the 0/1-error of the logistic regression estimate is bounded by*

$$P\left[\text{sign}(\hat{\boldsymbol{w}}^T \boldsymbol{x}) \neq y\right] \leq e^{-2k(1/2-p)^2} + \frac{1}{\sqrt{N}}\left(8BR + 12\sqrt{\log(2/\delta)/2}\right). \tag{4}$$

The proof of the proposition is in Appendix M.1. The first term in Equation 4 is (a bound on) the error of the Bayes-optimal estimator with $kn$ parameters. As the number of parameters increases (from $k$ up to $m$) the error decreases. The second term is the error induced by learning this estimator based on $N$ examples. The behavior in Figure 6 is consistent with such a bound. Once the model is sufficiently large, the first term in the bound is close to zero and does not significantly improve further by increasing the model size. The second term describes a power law in the number of training examples, which is what we also observe empirically.

## 6.2 Transformer analysis for substitution-only reconstruction

We now extend the analysis from linear models to transformers and from binary to the quaternary alphabet. Under i.i.d. substitution errors with rate $p_s < 0.25$, uniform sequence priors, and independent traces, the Bayes-optimal estimator that minimizes cross-entropy loss reduces to majority voting, which selects the base that appears most frequently at each position across traces (see also Appendix M.2).

**Theorem 1.** *There exists a 2-layer transformer with hidden dimension $d = |\mathcal{V}| + L$, where $|\mathcal{V}|$ is the vocabulary size and $L$ is the sequence length, that implements majority voting for trace reconstruction.*

The construction is given in Appendix M.3.

**Theorem 2.** *Let $f_\theta$ be any estimator achieving cross-entropy loss $\mathcal{L}(\theta) \leq \mathcal{L}^* + \delta$, where $\mathcal{L}^*$ is the Bayes-optimal loss. Then*

$$\mathbb{E}\left[\|P_\theta[\cdot] - P_{maj}[\cdot]\|_{TV}\right] \leq \sqrt{\tfrac{\delta}{2}},$$

*where $P_\theta[\cdot]$ and $P_{maj}[\cdot]$ denote the output distributions of $f_\theta$ and the Bayes-optimal estimator.*

This result, which we prove in Appendix M.4, implies that any estimator with near-optimal loss must make predictions close to those of majority voting. While convergence of gradient descent remains an open theoretical question, experiments in Appendix M.5 show that transformers trained with gradient descent achieve $\delta < 2 \times 10^{-4}$, and prediction confidence strongly correlates with vote margin.

## 7 Conclusion and limitations

In this work, we proposed a deep-learning-based approach for trace reconstruction and validated it in the context of DNA data storage. Our method, TReconLM, achieves lower Levenshtein distances and failure rates than state-of-the-art methods across small cluster sizes and a wide range of noise levels on both synthetic and real-world data.

Such a learning-based approach to trace reconstruction is suitable whenever training data can be simulated, which is often the case, also beyond DNA data storage.

The main limitation of TReconLM over classical, non-deep-learning based methods is that it requires training data and can perform poorly if test data is very different from the training data.

## Acknowledgments

Funded by the European Union (DiDAX, 101115134). Views and opinions expressed are however those of the author(s) only and do not necessarily reflect those of the European Union or the European Research Council Executive Agency. Neither the European Union nor the granting authority can be held responsible for them.

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

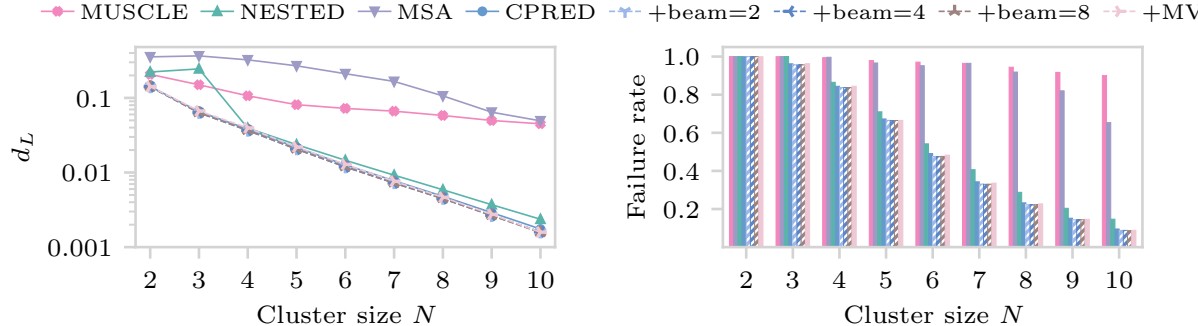

Figure 7: Comparison of different prediction targets and decoding strategies. The candidate prediction target (CPRED) achieves the lowest Levenshtein distances and failure rates. CPRED plus beam search or permutation-based majority voting (+MV) provides minimal improvement over greedy decoding at increased inference cost.

## A   Multiple sequence alignment target

In this section, we evaluate different prediction targets for trace reconstruction. As proposed by Dotan et al. (2023), we can train a model $f_{\boldsymbol{\theta}}$ to learn the alignment of traces. For $N$ traces $\boldsymbol{y}_1, \ldots, \boldsymbol{y}_N$, one training instance is formed as

$$\boldsymbol{y}_1 \mid \boldsymbol{y}_2 \mid \ldots \mid \boldsymbol{y}_{N-1} \mid \boldsymbol{y}_N : \mathrm{MSA}\big(\boldsymbol{y}_1, \boldsymbol{y}_2, \ldots, \boldsymbol{y}_{N-1}, \boldsymbol{y}_N\big)\texttt{\#}. \tag{5}$$

The vocabulary for the alignment task is given by

$$\mathcal{V}_{\mathrm{MSA}} = \{\mathrm{A}, \mathrm{C}, \mathrm{G}, \mathrm{T}, \mathtt{|}, \mathtt{:}, \mathtt{-}, \texttt{\#}\}, \tag{6}$$

where we have an additional deletion token $\mathtt{-}$ to achieve a column-wise alignment of the traces. For pretraining, we know the positions of insertions, deletions, and substitutions and can construct the correct sequence alignment. During inference, we prompt the model with input $\boldsymbol{p}$ defined in Equation 1 and generate alignment tokens one by one until the padding token $\texttt{\#}$.

To obtain a sequence estimate $\hat{\boldsymbol{x}}$, we arrange the aligned traces $\hat{\boldsymbol{y}}_1, \ldots, \hat{\boldsymbol{y}}_N$, each of length $L_{\mathrm{MSA}}$, as rows in a matrix:

$$
\begin{array}{|c|c|c|c|c|}
\hline
\hat{y}_{1,1} & \hat{y}_{1,2} & \cdots\cdots & \hat{y}_{1,L_{\mathrm{MSA}}-1} & \hat{y}_{1,L_{\mathrm{MSA}}} \\
\hline
\hat{y}_{2,1} & \hat{y}_{2,2} & \cdots\cdots & \hat{y}_{2,L_{\mathrm{MSA}}-1} & \hat{y}_{2,L_{\mathrm{MSA}}} \\
\hline
\vdots & & \vdots & & \vdots \\
\hline
\hat{y}_{N,1} & \hat{y}_{N,2} & \cdots\cdots & \hat{y}_{N,L_{\mathrm{MSA}}-1} & \hat{y}_{N,L_{\mathrm{MSA}}} \cdot \\
\hline
\end{array}
\tag{7}
$$

We then compute $\hat{\boldsymbol{x}}$ by performing a column-wise majority vote over the aligned traces. The $j$-th entry of the estimated sequence $\hat{\boldsymbol{x}}$ can be calculated as

$$\hat{x}_j = \operatorname*{arg\,max}_{a \in \{\mathrm{A}, \mathrm{C}, \mathrm{G}, \mathrm{T}\}} \sum_{i=1}^{N} \mathbf{1}(\hat{y}_{i,j} = a), \tag{8}$$

where $\mathbf{1}(\cdot)$ denotes the indicator function.

We evaluate the following targets for trace reconstruction: candidate prediction (CPRED) as described in Section 4, the MSA target as given in Equation 5, and a NESTED alignment target, where we perform a token-wise nesting of the ground-truth alignment $\mathrm{MSA}(\boldsymbol{y}_1, \ldots, \boldsymbol{y}_N)$. All deep-learning models are trained under a fixed compute budget of $1.0 \times 10^{20}$ FLOPs. We also evaluate MUSCLE to compare neural network-based alignment to dynamic programming-based alignment.

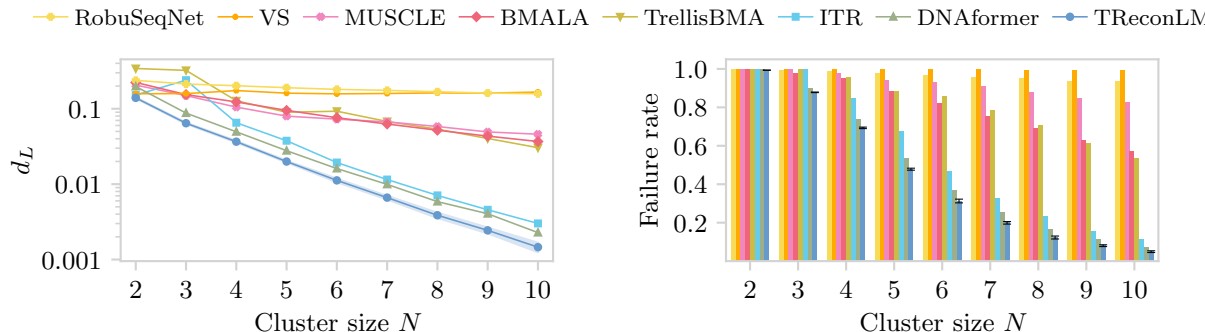

Figure 8: Average Levenshtein distances $d_L$ and failure rates on synthetic data with sequence length $L = 60$. TReconLM is averaged over three runs with different seeds. Shaded bands and error bars show $\pm$ one standard deviation.

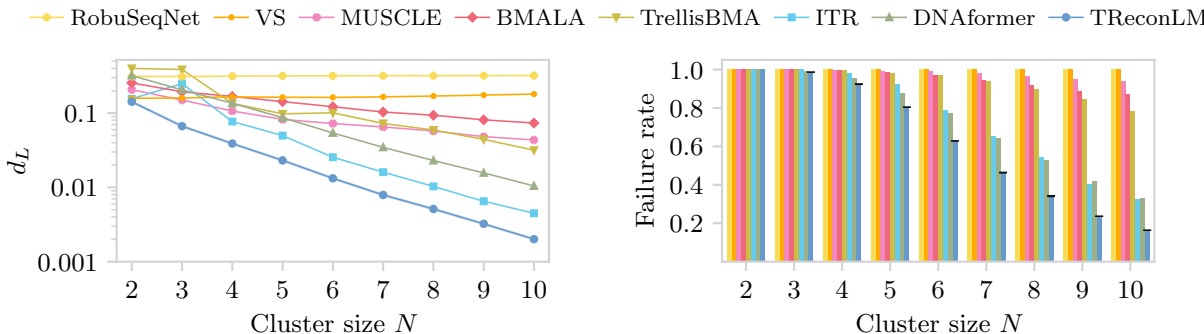

Figure 9: Average Levenshtein distances $d_L$ and failure rates on synthetic data with sequence length $L = 180$. TReconLM is averaged over three runs with different seeds. Shaded bands and error bars show $\pm$ one standard deviation, which is very small and hardly visible.

Figure 7 shows Levenshtein distances $d_L$ for all target types. The CPRED target achieves the best overall performance. In contrast, alignment-based targets require longer context lengths and cannot be used for fine-tuning, as ground-truth alignments are generally unavailable for real-world data.

## B    Decoding strategies

In this section, we test different decoding strategies for TReconLM trained on synthetic data with sequence length $L = 110$. We compare greedy decoding against beam search and against permutation-based majority voting, where we decode up to 10 random permutations of the input traces and take a per-position majority vote across the resulting reconstructions.

Figure 7 shows the results for the different decoding strategies. Beam search gives only marginal performance gains over greedy decoding at the cost of increased inference time, as the model's predictive distributions are very peaked (mean top-1/top-2 probabilities of 96.8%/2.6% on a 1K example subset). Permutation-based majority voting also provides minimal gains. Using the first permutation for tie-breaking, which outperforms random selection, failure rate improves by 0.5% but average Levenshtein distance increases by 0.03%. The reconstructions from different permutations differ by an average of 7.2 nucleotides (pairwise Hamming distance), with a per-position vote agreement of 0.96 (i.e., at each position, 96% of permutations agree on the same nucleotide).

To further quantify robustness to input ordering, we evaluate TReconLM 20 times on the test set, each time with a different random permutation of the input traces for every example, and report the mean and standard deviation across runs. The overall failure rate is $52.89\% \pm 0.52\%$ and the overall Levenshtein distance is

Table 1: Optimization hyperparameters used during pretraining and fine-tuning.

| Setting | Details | Batch | Context length | Iter.[*] | $n_{emb}$ | $n_{head}$ | $n_{layers}$ | Adam $\beta$ | Weight decay | LR | Dropout |
|---|---|---|---|---|---|---|---|---|---|---|---|
| Pretraining | $L = 60$ | 800 | 800 | 688,318 | 512 | 8 | 12 | (0.9, 0.95) | 0.1 | 7e-4 | 0.0 |
| | $L = 110$ | 800 | 1500 | 367,103 | 512 | 8 | 12 | (0.9, 0.95) | 0.1 | 7e-4 | 0.0 |
| | $L = 180$ | 800 | 2400 | 229,439 | 512 | 8 | 12 | (0.9, 0.95) | 0.1 | 7e-4 | 0.0 |
| Fixed $N$ | $N = 10$ | 800 | 1500 | 367,103 | 512 | 8 | 12 | (0.9, 0.95) | 0.1 | 7e-4 | 0.0 |
| | $N = 20$ | 512 | 2800 | 315,368 | 512 | 8 | 12 | (0.9, 0.95) | 0.1 | 1e-4 | 0.1 |
| | $N = 30$ | 256 | 4500 | 383,260 | 512 | 8 | 12 | (0.9, 0.95) | 0.1 | 1e-4 | 0.2 |
| | $N = 40$ | 256 | 5800 | 311,503 | 512 | 8 | 12 | (0.9, 0.95) | 0.1 | 1e-4 | 0.2 |
| | $N = 50$ | 256 | 6500 | 263,579 | 512 | 8 | 12 | (0.9, 0.95) | 0.1 | 1e-4 | 0.2 |
| Variable $L$ | $L \in [50, 120]$ | 400 | 2400 | 459,912 | 512 | 8 | 12 | (0.9, 0.95) | 0.1 | 1e-4 | 0.1 |
| Fine-tuning | Noisy-DNA | 8 | 800 | 685,307 [**] | 512 | 8 | 12 | (0.9, 0.95) | 0.1 | 1e-5 | 0.1 |
| | Microsoft | 25 | 1500 | 566,046 | 512 | 8 | 12 | (0.9, 0.95) | 0.001 | 1e-5 | 0.1 |
| | Chandak et al. | 8 | 2400 | 685,307[***] | 512 | 8 | 12 | (0.9, 0.95) | 0.1 | 1e-5 | 0.1 |

[*] Iterations are chosen to meet a fixed compute budget for each experiment.
[**] Early stopped after 165,000 iterations for total compute of $2.5 \times 10^{17}$. [***] Early stopped after 431,000 iterations for total compute of $1.9 \times 10^{18}$.

$0.0339 \pm 0.0003$, with small standard deviations indicating that TReconLM is robust to the ordering of input traces.

## C  Additional results on synthetic data and implementation details

In this section, we additionally evaluate TReconLM's performance on reconstructing sequences of length $L = 60$ and $L = 180$. We use the same model size of $\sim$38M parameters and the same compute budget of $1.0 \times 10^{20}$ FLOPs as in Section 5.2 of the main paper. Both models are trained on $\sim$440B tokens. The number of training examples is adjusted based on the context length to match the fixed compute budget, with $\sim$551M examples for $L = 60$ and $\sim$184M examples for $L = 180$.

Optimization hyperparameters are listed in Table 1, largely following Karpathy (2025). We apply gradient clipping with a maximum norm of 1.0. The learning rate is scaled based on batch size. The base learning rate is 1e-4 for batch size 16, and we scale it proportionally to $\sqrt{\text{batch size}/16}$. We use a 5% warmup phase followed by cosine learning rate decay. For pretraining with a fixed cluster size and for fine-tuning, we use fixed learning rates without scaling based on batch size. Unless stated otherwise, we evaluate the checkpoint with the lowest validation loss.

Figure 8 shows Levenshtein distances and failure rates for sequence length $L = 60$. Figure 9 shows the corresponding results for sequence length $L = 180$. For both lengths, TReconLM achieves lower Levenshtein distances and failure rates across all cluster sizes considered and outperforms the state-of-the-art ITR algorithm (Sabary et al., 2024) as well as other neural approaches (Bar-Lev et al., 2025; Qin et al., 2024).

## D  Failure modes and attention patterns

In this section, we analyze failure modes to identify what makes sequences difficult for TReconLM to reconstruct and visualize attention matrices across layers. We define the overall error rate $r_{overall}$ as total reconstruction errors divided by total nucleotides in the test set. Reconstruction errors are computed as the minimal number of edit operations (insertions, deletions, and substitutions) between predicted and ground-truth sequences.

**Sequence pattern analysis.** To test whether local sequence context influences reconstruction difficulty, we analyze the synthetic test set with sequence length $L = 110$, which has uniform error rates without position-dependent or technology-specific biases. For each subsequence of length 3, 5, or 7, we define $N_{pattern}$ as the total number of times the pattern appears across ground-truth sequences and $E_{pattern}$ as the total reconstruction errors at those positions. Under the null hypothesis of uniform errors, $E_{pattern}$ follows a Binomial$(N_{pattern}, r_{overall})$ distribution. We apply two-sided binomial tests to identify subsequences with error rates significantly different from $r_{overall}$ (testing subsequences with $N_{pattern} \geq 20$) and use Benjamini-Hochberg correction (Benjamini & Hochberg, 1995) to control false discovery rate at 5%.

Table 2: Top 5 sequence patterns with significantly higher reconstruction error rates than expected under a uniform error distribution.

| Length 3 | | | Length 5 | | | Length 7 | | |
|---|---|---|---|---|---|---|---|---|
| Pattern | $N_{\text{pattern}}$ | Enrichment | Pattern | $N_{\text{pattern}}$ | Enrichment | Pattern | $N_{\text{pattern}}$ | Enrichment |
| CTT | 84511 | 1.58x | ACAAA | 5247 | 2.33x | ACTCCCC | 306 | 3.82x |
| GCC | 84619 | 1.54x | GAGGG | 5228 | 2.27x | CGTGGGG | 306 | 3.82x |
| ATT | 84317 | 1.54x | GTGGG | 5317 | 2.23x | CTTGGGG | 297 | 3.73x |
| TCC | 84119 | 1.51x | TCTTT | 5201 | 2.21x | CATGGGG | 303 | 3.66x |
| GAA | 83857 | 1.51x | TTCCC | 5148 | 2.21x | AAATTTT | 305 | 3.63x |

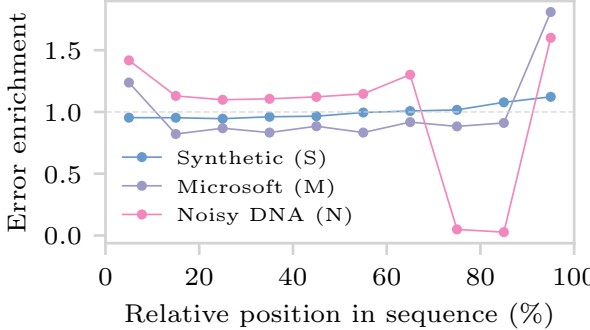 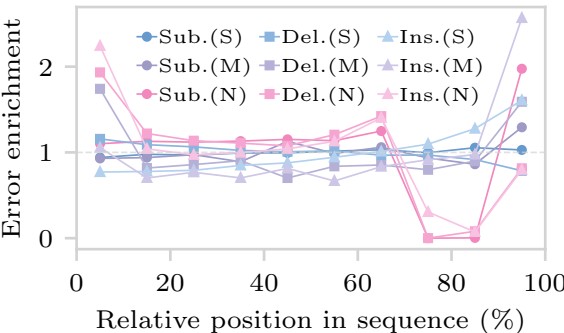

Figure 10: Positional biases in reconstruction errors for synthetic and real-world datasets. Left: average across all error types. Right: breakdown by error type.

Table 2 shows the top 5 patterns with significantly higher error rates for each length (adjusted $p < 0.05$). Longer patterns show greater reconstruction difficulty, with enrichment (the ratio of observed to expected error rates) up to 3.82 for length 7. The most difficult patterns across all lengths contain consecutive identical nucleotides (e.g., CTT, ACAAA, ACTCCCC).

To test whether runs, which we define as three or more consecutive identical nucleotides, are more difficult to reconstruct, we classify each edit operation by whether it affects a run. Similar to the pattern analysis, we use a binomial test to determine if the error rate in runs is significantly higher than $r_{\text{overall}}$. We find that runs make up 15.4% of sequence positions but account for 21.6% of reconstruction errors (enrichment 1.41, $p < 0.001$). The primary failure mode is run interruption (37.6% of run errors), where the model breaks up runs by substituting incorrect nucleotides, followed by boundary errors at run edges (29.7%), contractions where the model predicts shorter run lengths (23.6%), and expansions where the model predicts longer run lengths (9.1%).

**Positional error analysis.** We next identify positional biases in reconstruction errors. Figure 10 shows total error enrichment (ratio of observed to expected error rates, left) and error type breakdown (right). The synthetic test set shows no positional bias, as expected from a uniform error distribution.

For the Microsoft dataset, deletions occur more frequently in the first and last 10% of sequence positions (enrichment 1.74 and 1.57, respectively), insertions occur 2.57 times more frequently in the final 10%, and substitutions are approximately uniform with only a slight increase in the final 10% (enrichment 1.32).

For the Noisy-DNA dataset, deletions and insertions occur more frequently at sequence starts (enrichment 1.94 and 2.23 in the first 10%, respectively), while substitutions occur more frequently at sequence ends (enrichment 1.97 in the final 10%). Positions 70-90% show near-zero error rates due to the index sequence at positions 42-54. We cluster reads by index and discard traces with index errors, resulting in identical indices across all traces in each cluster. Additionally, the first 5 nucleotides of every index are the fixed value

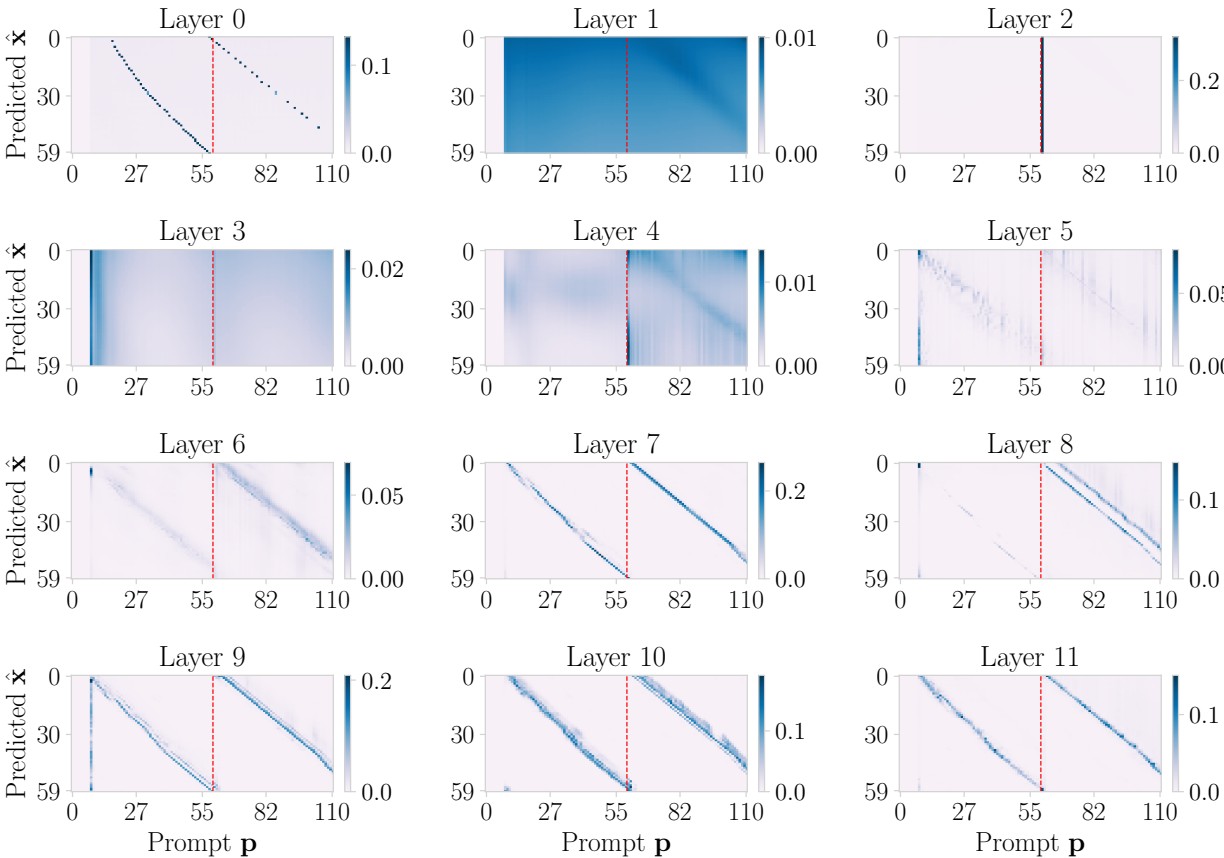

Figure 11: Min-max normalized attention maps for all layers of TReconLM on a representative synthetic example (cluster size 2, sequence length L=60). The vertical dashed lines mark the trace boundaries.

`ACAAC`. The model learns this fixed prefix as a synchronization marker, allowing it to align noisy traces with ground-truth positions and reconstruct the entire index region with near-zero errors.

**Attention patterns.** We visualize attention maps across layers in Figure 11. Early layers show different structures, with Layer 0 showing thin diagonals that may reflect an initial guess of how parts of the traces align with the output sequence, Layer 1 attending broadly across almost all positions, and Layer 2 focusing almost entirely on the separator tokens. From Layer 3 onward, the attention maps show broader diagonals that become sharper in deeper layers, indicating that the model progressively refines the alignment. We find that this overall pattern is consistent across different examples and cluster sizes.

# E    Robustness to misclustered traces

This section provides further details on how clustering errors affect TReconLM's reconstruction performance for synthetic data with sequence length $L = 110$.

Figure 12 shows how misclustering affects model performance across different cluster sizes. We measure Levenshtein distance increases compared to the same model evaluated on the test set with no misclustering. Larger clusters ($N > 5$) are robust, showing minimal Levenshtein distance increases across all misclustering rates $p_m$. This holds even in the worst-case setting, where the model is trained on perfectly clustered data but evaluated only on test clusters with at least one misclustered trace.

For smaller cluster sizes ($3 \leq N \leq 5$), training on misclustered data leads to moderate performance improvements. For example, at cluster size 3 and $p_m = 0.02$, when evaluated only on test clusters with at least one misclustered trace, the $d_L$ increase drops from 0.17 (clean training) to 0.15 (misclustered training).

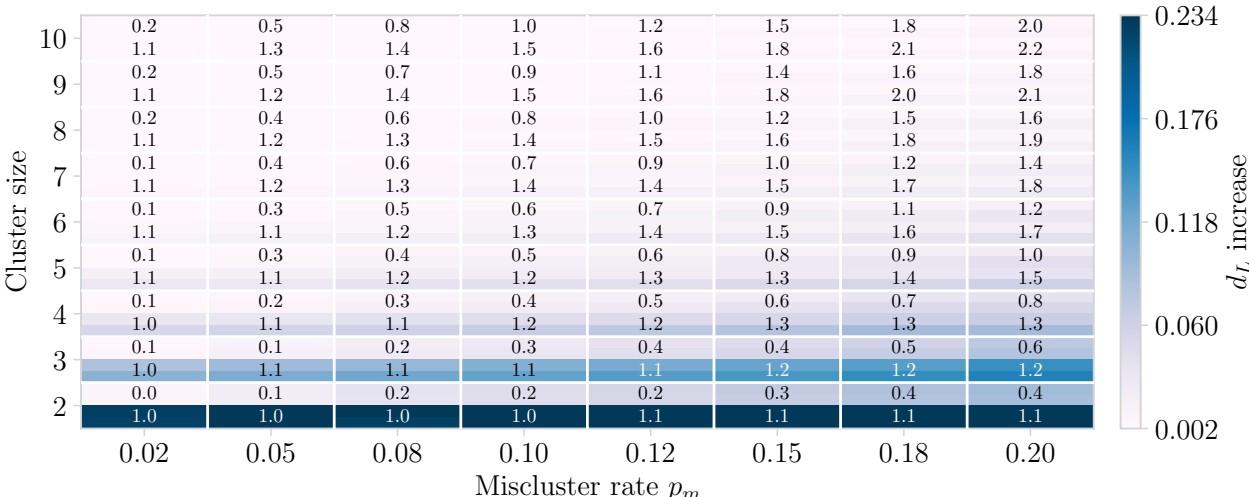

Figure 12: Levenshtein distance $d_L$ increase across cluster sizes and misclustering rates $p_m$. For each cluster size and misclustering rate, the bottom half shows $d_L$ increase for examples with at least one misclustered trace and the top half for all examples. Numbers indicate average misclustered sequences per cluster, respectively. Within each half, the lower row shows the model trained on perfectly clustered data and the upper row shows the model trained with misclustered data.

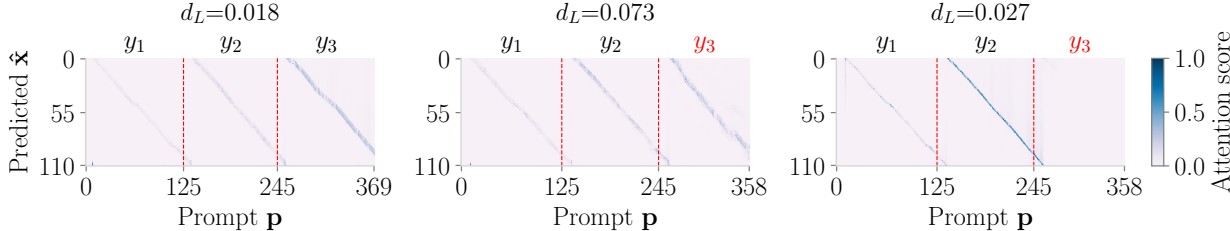

Figure 13: Min-max normalized attention maps for a cluster of size 3 with one misclustered trace (red). Left: trained on clean, evaluated on clean. Middle: trained on clean, evaluated on misclustered. Right: trained on misclustered, evaluated on misclustered.

Performance degrades most at cluster size 2, with a $d_L$ increase of 0.234. With an average of one misclustered trace per cluster, the model cannot distinguish between correct and incorrect sequences without ground-truth, resulting in similar performance loss regardless of training conditions.

We find that the Levenshtein distance between misclustered sequences and the ground-truth has no impact on reconstruction performance. This is expected since misclustered sequences come from different random sequences and thus have large Levenshtein distances from the ground-truth by construction.

To understand why training on misclustered data improves robustness, we visualize attention matrices from the last layer for prompts with cluster size 3 in Figure 13. The attention shows a diagonal structure where read position $j$ attends to sequence estimate position $j$. Comparing models trained on clean versus misclustered data shows that the model trained on misclustered data learns to ignore misclustered traces, while the model trained on clean data attends to all sequences.

## F    Pretraining ablation

To assess the effect of pretraining, we train TReconLM from scratch on the Noisy-DNA and Microsoft datasets, matching the compute budget and hyperparameters of the pretraining runs. Table 3 shows average

Table 3: Effect of pretraining on the Noisy-DNA and Microsoft datasets. Columns (p.) report performance with pretraining.

| Dataset | Average $d_L$ | Average $d_L$ (p.) | Failure rate | Failure rate (p.) |
|---|---|---|---|---|
| Noisy-DNA | 0.259 | **0.239** | 0.903 | **0.834** |
| Microsoft | 0.014 | **0.009** | 0.342 | **0.205** |

Table 4: Fine-tuning data efficiency on the Noisy-DNA dataset. Both metrics improve as the fraction of fine-tuning data increases.

| Metric | 0% | 5% | 10% | 25% | 50% | 75% | 100% |
|---|---|---|---|---|---|---|---|
| Levenshtein distance | 0.307 | 0.267 | 0.259 | 0.252 | 0.242 | 0.240 | **0.236** |
| Failure rate | 0.999 | 0.887 | 0.866 | 0.851 | 0.840 | 0.834 | **0.831** |

Levenshtein distances and failure rates across cluster sizes, showing that pretraining improves performance on both datasets.

## G    Noisy-DNA dataset preprocessing details

This section describes the preprocessing steps for the Noisy-DNA dataset experiment in Section 5.3.1.

For validation and test sets, we precompute fixed subclusters by repeatedly sampling a cluster size between 2 and 10 and selecting that many traces without replacement. We stop when fewer than two traces remain and discard them. During training, we apply the same subclustering strategy but sample only one subcluster per example in each epoch, using the 13,104 training examples whose cluster size can exceed 10. This dynamic subclustering increases training diversity by showing the model different random subsets of traces (for large clusters) and different trace orderings (across all clusters), effectively augmenting the data.

Given a sequence length $L = 60$ and estimated error probabilities $p_I = 0.057$, $p_D = 0.06$, and $p_S = 0.026$, the expected number of edit operations per trace is $L \times (p_I + p_D + p_S) = 8.58$. We set a conservative threshold and remove all traces from the train (30,546 of 690,395) and validation (3,905 of 87,429) sets whose Levenshtein distance to any test-set ground-truth sequence falls between 5 and 13. We further discard any clusters with fewer than two remaining reads (2 in the training set, 1 in the validation set).

For the non-deep-learning baselines and our pretrained TReconLM model, we perform an additional pre-processing step on the test set that removes trailing C nucleotides to improve performance. During library preparation, a C-rich tail is artificially added to each sequence for chemical reasons. Under normal conditions, this tail remains outside the 60-base sequencing window. However, a high deletion rate during synthesis can shift the C-rich tail into the sequencing window. For fine-tuning, we do not apply this preprocessing step, to allow the model to adapt to the dataset's error characteristics.

## H    Fine-tuning data efficiency

We analyze how sensitive TReconLM's performance is to the amount of fine-tuning data. We fine-tune the pretrained model (sequence length $L = 60$) on different fractions of the Noisy-DNA training set (5%, 10%, 25%, 50%, 75%), and compare against both the pretrained model (0%) and the fully fine-tuned model (100%). We consider the Noisy-DNA dataset because it shows a large gap between pretrained and fine-tuned performance, whereas on the Microsoft dataset the pretrained model already performs well, so we expect less sensitivity to the amount of fine-tuning data.

Table 4 reports Levenshtein distance and failure rate (averaged across cluster sizes). Both metrics improve as the fraction of fine-tuning data increases, with the largest gains observed when training on the full dataset.

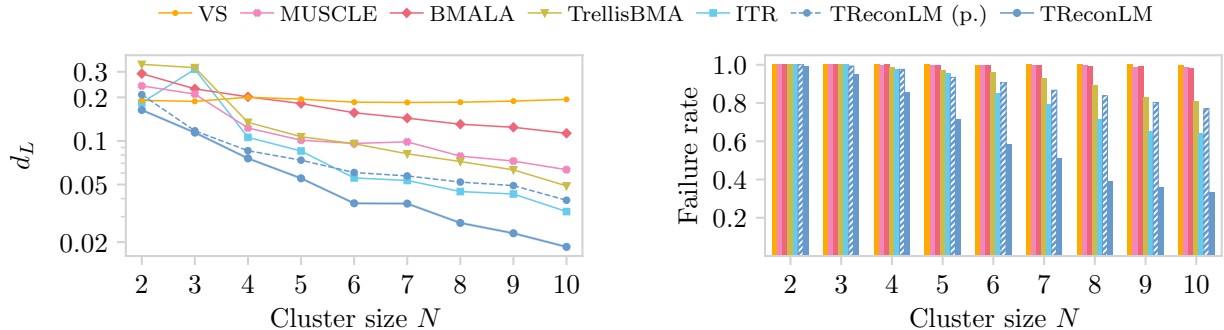

Figure 14: Average Levenshtein distances $d_L$ and failure rates on the Chandak et al. (2020) dataset for variable-length pretrained TReconLM (p.), fine-tuned TReconLM, and classical baselines.

## I  Additional results on real-world data

In this section, we evaluate the variable-length model from Section 5.2.4 (pretrained on sequences of length $L \in [50, 120]$) on an additional real-world dataset from Chandak et al. (2020). This dataset uses nanopore sequencing with estimated error rates of approximately 10%, roughly equally distributed across insertions, deletions, and substitutions.

The dataset consists of 13 experiments with ground-truth sequences of lengths between 108 and 119 nucleotides and a 25-nucleotide primer at each end for experiment identification. We group the noisy traces by experiment, searching for the forward primer in the first and the reverse primer in the second half of each trace and testing both orientations. We assign each trace to the experiment whose primer pair has the smallest combined edit distance. Of the 3,444,000 total traces, we exclude 7 that are too short to contain both primers.

We select 6 of the 13 experiments that have a uniform ground-truth sequence length of 117 nucleotides. We use experiments 4, 8, 9, and 12 for training (3,115 ground-truth sequences total, 832,698 noisy reads), experiment 10 for validation (877 ground-truth sequences, 122,106 noisy reads), and experiment 11 for testing (815 ground-truth sequences, 131,086 noisy reads). Similar to the Noisy-DNA dataset, we cluster reads by index using the shortest unique prefix (10 nucleotides) that uniquely identifies each ground-truth sequence, and discard reads with index errors.

After clustering, we filter reads by length, removing those shorter than 100 or longer than 140 nucleotides, and construct subclusters using the same approach as described for the Noisy-DNA and Microsoft datasets in Section 5.3. For validation and test sets, we precompute fixed subclusters by repeatedly sampling subcluster sizes between 2 and 10, resulting in 8,212 validation and 8,506 test examples. For the training set, we apply dynamic subclustering, sampling min(cluster size, 10) traces from the 3,115 clusters in random order each epoch to increase data diversity.

Figure 14 shows average Levenshtein distances and failure rates for each cluster size. We compare our variable-length pretrained TReconLM, TReconLM fine-tuned on the dataset from Chandak et al. (2020), and classical baselines. We fine-tune with a compute budget of $1.9 \times 10^{18}$ FLOPs for 431,000 iterations (stopping early before the full 685,307 iterations). Our fine-tuned TReconLM reconstructs 19.36% more sequences without error compared to the state-of-the-art ITR algorithm, demonstrating that the variable-length model generalizes effectively when fine-tuned on fixed-length real-world data.

## J  Additional baseline comparisons

Here, we provide additional comparisons to RobuSeqNet, DNAformer, GPT-4o mini, and GPT-5.

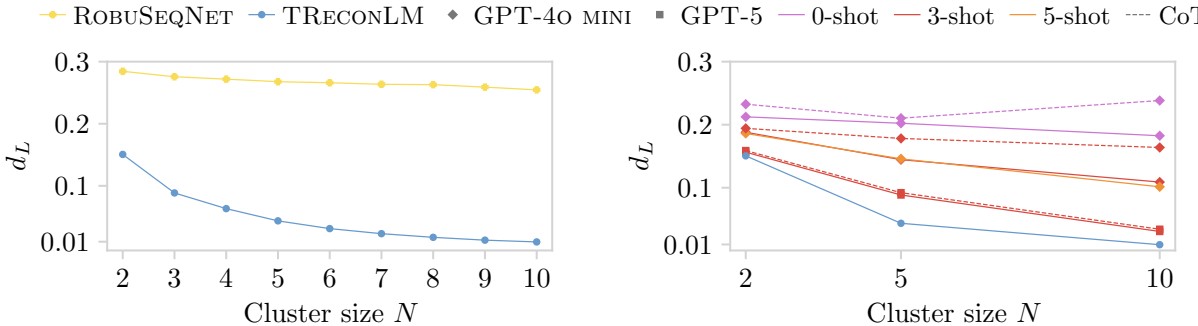

Figure 15: Left: Comparison of TReconLM and RobuSeqNet at equal model size and compute for ground-truth sequence length $L = 110$ and cluster sizes $N \in [2, 10]$. Right: Average Levenshtein distance $d_L$ for GPT-4o mini, GPT-5, and TReconLM on trace reconstruction with sequence length $L = 60$, using two, five, or ten noisy reads. GPT-4o mini and GPT-5 are evaluated with 0-, 3-, and 5-shot prompting, with and without Chain-of-Thought (CoT).

### J.1  RobuSeqNet

We compare the performance of TReconLM to RobuSeqNet (Qin et al., 2024) when controlling for model size and compute. RobuSeqNet is a small model with $\sim$3M parameters and uses an LSTM decoder with a hidden dimension of 256. We train a TReconLM model with the same hidden dimension and total parameter count ($\sim$3M), using the same compute budget ($6 \times 10^{17}$ FLOPs) and training dataset ($\sim$21M examples).

Figure 15, left panel, shows results for sequence length $L = 110$, evaluated on 50K test examples (identical to the test set used in Figure 2 of the main paper). TReconLM also outperforms RobuSeqNet when controlling for model size and compute.

### J.2  DNAformer

We compare TReconLM to the self-reported performance of DNAformer (Bar-Lev et al., 2025), which was trained on synthetic data and evaluated on the Microsoft dataset with up to 16 reads per example. For a fair comparison, we evaluate our pretrained TReconLM (input length $L = 110$) and recluster the noisy reads by index, as in Bar-Lev et al. (2025). Since the Microsoft dataset does not have explicit indices, we follow their approach and use the shortest unique prefix of each ground-truth sequence as the index. We then cap each cluster at a maximum of 10 reads to match TReconLM's context length. This results in 9,729 test examples.

Bar-Lev et al. (2025) report a failure rate of 0.146 for DNAformer, whereas TReconLM achieves 0.111 with fewer reads and no dynamic-programming postprocessing. While DNAformer was trained on synthetic data generated using error statistics derived from the real dataset, TReconLM was pretrained on a fixed noise distribution and was not tuned to the Microsoft dataset. Thus, TReconLM performs better under worse initial conditions.

### J.3  GPT-4o mini and GPT-5

To test whether general-purpose LLMs can perform trace reconstruction via in-context learning and reasoning, we include GPT-4o mini and GPT-5 as additional baselines.

We prompt both models as shown in Figure 18 to reconstruct sequences of length $L = 60$ using zero-, three-, and five-shot prompting. For TReconLM, we use a 3M-parameter model with the same architecture as in Section J, trained on $\sim$39.5M examples with a compute budget of $6 \times 10^{17}$ FLOPs. The training set used here is larger than in Section J because of the shorter target length ($L = 60$ vs. 110).

We evaluate GPT-4o mini on 250 synthetic test instances per cluster size (2, 5, and 10), and GPT-5 on 100 instances for cost reasons. All examples use error probabilities drawn uniformly from $[0.01, 0.1]$. The

few-shot examples shown to GPT-4o mini and GPT-5 are sampled from the same distribution as the test set.

Figure 15, right panel, shows that TReconLM achieves lower Levenshtein distances than GPT-4o mini and GPT-5 across all tested cluster sizes. We additionally evaluate both models with Chain-of-Thought prompting by adding an instruction to "first think step by step about how the input traces align and which positions are reliable" before providing the reconstruction, and find that Chain-of-Thought does not improve reconstruction accuracy for either model.

## K  Baseline methods

Here, we describe the implementation details and hyperparameters for all baseline methods used in our experiments.

### K.1  Non-deep-learning method parameters

We evaluate each non-deep-learning baseline using the parameters specified in the original publications. When error probabilities $p_I, p_D, p_S$ are required, we use estimates for the real datasets (Antkowiak et al., 2020; Srinivasavaradhan et al., 2021), and the average values of the corresponding noise distributions for synthetic data, except for TrellisBMA at increased noise levels (Section 5.2.1). For TrellisBMA, we find that using the true average values at higher noise levels leads to noticeably worse performance. Instead, we fix the error parameters to the average of the base noise distribution (corresponding to $k = 0$).

For BMALA and VS, we adopt the parameters from Sabary et al. (2024). BMALA uses a window size of $w = 3$. For the VS algorithm, we compute $\delta = (1 + p_S)/2$ and set $\gamma = 3/4$, $r = 2$, and $l = 5$.

For TrellisBMA, we use the same parameters as in Srinivasavaradhan et al. (2021), setting $\beta_b = 0$ for all cluster sizes $N$ from 2 to 10, and adapting $\beta_e$ and $\beta_i$ based on the cluster size. For cluster sizes $N \in \{2, 3\}$, we use $(\beta_e, \beta_i) = (0.1, 0.5)$; for cluster sizes $N \in \{4, 5\}$, $(1.0, 0.1)$; for cluster sizes $N \in \{6, 7\}$, $(0.5, 0.1)$; for cluster sizes $N \in \{8, 9\}$, $(0.5, 0.5)$; and for cluster size $N = 10$, $(0.5, 0.0)$.

### K.2  Deep-learning baselines

We first briefly describe the two deep-learning baselines we consider, RobuSeqNet (Qin et al., 2024) and DNAformer (Bar-Lev et al., 2025), and then list the hyperparameters used in our experiments.

#### K.2.1  RobuSeqNet

RobuSeqNet takes clusters of one-hot encoded, padded DNA reads as input and outputs per-position nucleotide predictions for the reconstructed sequence. The model consists of five main blocks:

- **Read-weighting module:** Computes a weight for each read (from a convolutional feature representation) and multiplies this weight with the original one-hot-encoded read. Weighted reads are summed to produce a single consensus sequence.

- **Linear projection module:** Projects the combined representation through a linear layer to map from the noisy input length to the target label length.

- **Convolutional upsampling module:** A 2D convolutional module that increases the feature size.

- **Conformer block:** Combines self-attention, depthwise convolutions, and feed-forward layers to update the feature representation.

- **RNN output module:** A two-layer LSTM processes the sequence representation and outputs per-position logits over the four nucleotides via a final linear layer.

**Training setup.** We adapt the original implementation to dynamically generate synthetic data, using the same noise distribution as in TReconLM pretraining. Because our data loader does not rely on a fixed dataset, we train for a single epoch with cosine learning rate decay and 5% linear warm-up.

We increase the pretraining batch size to 1500 for $L = 60$, 800 for $L = 110$, and 600 for $L = 180$ to match larger compute budgets, using maximum learning rates of $\mathrm{lr_{max}} = 6.1 \times 10^{-4}$, $7.1 \times 10^{-4}$, and $9.7 \times 10^{-4}$, respectively. This configuration performs slightly better than the default batch size of 64 used in the original implementation. For fine-tuning, we use batch sizes of 8 (Noisy-DNA) and 52 (Microsoft), with a maximum learning rate of $1 \times 10^{-5}$. All other hyperparameters follow the original implementation (Qin et al., 2024). Dropout is set to 0.1 for convolutional, RNN, and conformer-attention layers, and training uses Adam with $\beta = (0.9, 0.98)$.

### K.2.2 DNAformer

DNAformer likewise takes clusters of one-hot encoded, padded DNA reads as input and outputs per-position nucleotide predictions. It consists of five main modules:

- **Alignment module:** Learns a per-read alignment representation using four convolutional blocks with kernel sizes (1, 3, 5, 7) along the sequence dimension. The extracted features are concatenated and passed through a feed-forward block.

- **Embedding module:** Merges aligned read features into a single cluster representation by summing over the read dimension, followed by convolutional blocks and a feed-forward projection to the target label length.

- **Transformer encoder:** Uses multi-head self-attention to model dependencies across sequence positions and outputs updated embeddings.

- **Output module:** Maps the Transformer output to nucleotide logits using three 1D convolution layers.

- **Fusion module:** Each cluster is processed twice (original and reversed order) through shared-weight modules. The two logits sequences are combined position-wise using learned weights to produce the final sequence estimate.

**Training setup.** As with RobuSeqNet, we adapt the data loader to dynamically generate synthetic data using the same noise distribution as in TReconLM pretraining. We train for a single epoch with cosine learning rate decay and 5% linear warm-up. We follow the optimization hyperparameters from the original implementation, using the Adam optimizer with $\mathrm{lr_{max}} = 3 \times 10^{-5}$, $\mathrm{lr_{min}} = 1 \times 10^{-7}$, batch size 64, $\beta = (0.9, 0.999)$, and no dropout or weight decay. We also test larger batch sizes with scaled learning rates but observe slightly worse performance. For fine-tuning, we use batch sizes of 8 (Noisy-DNA) and 52 (Microsoft), with a maximum learning rate $1 \times 10^{-5}$. Dropout is set to 0.1 for Noisy-DNA and 0 for Microsoft.

## L Inference cost

In this section, we compare the inference time of TReconLM with all baselines. We report both throughput per hour and per dollar (throughput divided by hardware cost) to allow a fair comparison between CPU- and GPU-based methods.

We use Azure pricing from East US[1], where one A100 GPU costs approximately \$3.40/hour and 64 CPUs (E64a v4) cost approximately \$4.00/hour. For GPU-based methods, we measure throughput on one A100 GPU with a batch size of 400 (for 100% GPU utilization), averaging over 6 independent 5-minute runs (1 warmup, 5 measured) and extrapolating to throughput per hour. For CPU-based methods, we run the same experiments on an AMD EPYC 7642 48-Core processor (comparable to E64a v4) with 64 parallel workers.

---

[1]https://azure.microsoft.com/en-us/pricing/details/machine-learning/

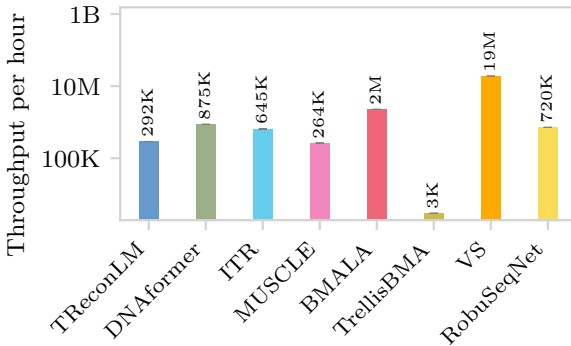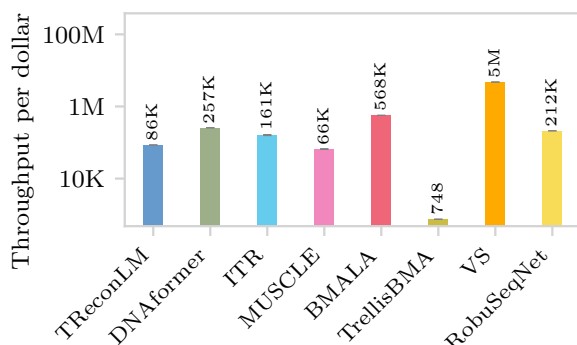

Figure 16: Throughput per dollar (solid) and per hour (transparent) for TReconLM and all baselines (higher is better). GPU-based methods use one A100 GPU ($3.40/hour) and CPU-based methods use 64 cores ($4.00/hour).

Figure 16 shows throughput per hour and dollar for all methods. TReconLM achieves comparable throughput per dollar to MUSCLE, which is widely used in practice for sequence alignment. While TReconLM's autoregressive generation requires multiple forward passes, making it slower than DNAformer and RobuSeqNet, which predict the entire sequence in a single forward pass, it achieves higher reconstruction accuracy. If inference speed becomes critical, transformer acceleration techniques could further improve TReconLM's throughput.

## M  Proofs for Section 6

Here we provide the proofs for the theoretical results in Section 6 and empirically validate how transformers solve trace reconstruction under substitution errors.

### M.1  Proof of Proposition 1

The proof of Proposition 1 is relatively standard.

The logistic (population) risk is

$$R(\boldsymbol{w}) = \mathbb{E}\left[\ell(\boldsymbol{w}^T\boldsymbol{x}, y)\right],$$

where $\ell(z, y) = \log(1 + e^{-yz})$ is the logistic loss. The empirical risk of the examples is defined in Equation 3.

From Bartlett et al. (2006), we have that the 0/1-excess loss for $\boldsymbol{w}$ is related to the logistic excess loss as follows. For any $\boldsymbol{w}$, we have that

$$\mathrm{P}\left[\mathrm{sign}(\boldsymbol{w}^T\boldsymbol{x}) \neq y\right] - \mathrm{P}\left[\mathrm{sign}(\boldsymbol{w}_B^T\boldsymbol{x}) \neq y\right] \leq 2(R(\boldsymbol{w}) - R(\boldsymbol{w}_*)). \tag{9}$$

where $\mathrm{sign}(\boldsymbol{w}_B^T\boldsymbol{x})$ is the Bayes-optimal classifier and $\boldsymbol{w}_*$ is the optimal logistic classifier. Therefore, we have that

$$\mathrm{P}\left[\mathrm{sign}(\hat{\boldsymbol{w}}^T\boldsymbol{x}) \neq y\right] \leq \mathrm{P}\left[\mathrm{sign}(\boldsymbol{w}_B^T\boldsymbol{x}) \neq y\right] + 2(R(\hat{\boldsymbol{w}}) - R(\boldsymbol{w}_*)) \tag{10}$$

$$\leq e^{-2k(1/2-p)^2} + 4\left(2\frac{BR}{\sqrt{N}} + \sqrt{\frac{9\log(2/\delta)}{2N}}\right), \tag{11}$$

where the last inequality holds with probability at least $1-\delta$. Here, we used that the Bayes error probability is the probability that at least half of the entries were flipped and is bounded by

$$\mathrm{P}\left[\mathrm{sign}(\boldsymbol{w}_B^T\boldsymbol{x}) \neq y\right] = \sum_{b=\lceil k/2\rceil}^{k} \binom{k}{b} p^b(1-p)^{k-b} \leq e^{-2k(1/2-p)^2}.$$

Moreover, we used the following bound, proven below. With probability at least $1 - \delta$,

$$R(\hat{\boldsymbol{w}}) - R(\boldsymbol{w}_*) \leq 2\left(2\frac{BR}{\sqrt{N}} + \sqrt{\frac{9\log(2/\delta)}{2N}}\right). \tag{12}$$

It remains to prove Bound 12.

We consider the function class

$$\mathcal{G} = \left\{(\boldsymbol{x}, y) \mapsto \ell(\boldsymbol{w}^T\boldsymbol{x}, y) \mid \|\boldsymbol{w}\|_2 \leq B\right\}.$$

Thus, $z = y\boldsymbol{w}^T\boldsymbol{x}$ lies in the interval $[-BR, BR]$. Because of this bound, the logistic loss is bounded by

$$0 < \ell(z) = \log(1 + e^{-z}) \leq \log(1 + e^{BR}).$$

From a standard generalization bound based on the Rademacher complexity (Bartlett et al., 2006), we get, for 1-Lipschitz loss and for all $\boldsymbol{w}$ with $\|\boldsymbol{w}\|_2 \leq B$ that

$$R(\boldsymbol{w}) \leq \hat{R}(\boldsymbol{w}) + 2r_N(\mathcal{G}) + \sqrt{\frac{9\log(2/\delta)}{2N}}$$

$$\leq \hat{R}(\boldsymbol{w}) + 2\frac{BR}{\sqrt{N}} + \sqrt{\frac{9\log(2/\delta)}{2N}}, \tag{13}$$

where $r_N$ is the Rademacher complexity. For the second inequality, we used the bound $r_N(\mathcal{G}) \leq \frac{BR}{\sqrt{N}}$ on the Rademacher complexity of linear estimators.

We have that

$$R(\hat{\boldsymbol{w}}) = \hat{R}(\hat{\boldsymbol{w}}) + R(\hat{\boldsymbol{w}}) - \hat{R}(\hat{\boldsymbol{w}}) \tag{14}$$

$$\leq \hat{R}(\boldsymbol{w}_*) + R(\hat{\boldsymbol{w}}) - \hat{R}(\hat{\boldsymbol{w}}) \tag{15}$$

$$\leq R(\boldsymbol{w}_*) + 2\left(2\frac{BR}{\sqrt{N}} + \sqrt{\frac{9\log(2/\delta)}{2N}}\right), \tag{16}$$

where first inequality holds because $\hat{\boldsymbol{w}}$ minimizes the empirical risk, and therefore $\hat{R}(\hat{\boldsymbol{w}}) \leq \hat{R}(\boldsymbol{w}_*)$, and for the last equality, we applied the generalization Bound 13 twice, once to bound $\hat{R}(\boldsymbol{w}_*)$, and once to bound $R(\hat{\boldsymbol{w}}) - \hat{R}(\hat{\boldsymbol{w}})$. This concludes the proof of Bound 12.

### M.2 Optimality of majority voting

We show that under i.i.d. substitution errors with uniform sequence priors and independent traces, the Bayes-optimal estimator reduces to majority voting.

For each position $i$, the posterior factorizes as

$$\mathrm{P}\left[x_i \mid \{y_i^j\}_{j=1}^N\right] \propto \prod_{j=1}^N \mathrm{P}\left[y_i^j \mid x_i\right], \quad \text{where } \mathrm{P}\left[y_i^j \mid x_i = b\right] = \begin{cases} 1 - p_s & \text{if } y_i^j = b, \\ p_s/3 & \text{otherwise.} \end{cases}$$

Let $n_b$ denote the number of traces with base $b$ at position $i$. Then

$$\hat{x}_i = \arg\max_b (1 - p_s)^{n_b}\left(\frac{p_s}{3}\right)^{N-n_b}.$$

and taking the logarithm (which preserves the argmax) gives

$$\hat{x}_i = \arg\max_b n_b \cdot \log\left(\frac{3(1-p_s)}{p_s}\right).$$

For $p_s < 0.25$, the coefficient is positive, so $\hat{x}_i = \arg\max_b n_b$, which is the majority voting rule that selects the base that appears most frequently at position $i$ across all traces.

### M.3 Proof of Theorem 1

We construct a transformer that takes as input concatenated traces as in Equation 1 and outputs $x_i$ according to majority voting.

Each token is embedded as $\mathbf{h} \in \mathbb{R}^d$ where $d = |\mathcal{V}| + L$ with vocabulary size $|\mathcal{V}| = 7$ for $\mathcal{V} = \{A, C, G, T, |, :, \#\}$ and maximum sequence length $L$. We concatenate token embeddings (dimension $|\mathcal{V}|$), which are one-hot vectors $\mathbf{e}_{\text{token}} \in \mathbb{R}^{|\mathcal{V}|}$, with position embeddings (dimension $L$), which are one-hot vectors $\mathbf{e}_{\text{pos}} \in \mathbb{R}^L$.

For the $j$-th token in the concatenated input:

$$\mathbf{h}_j = [\mathbf{e}_{\text{token}}; \mathbf{e}_{\text{pos}}] = \begin{cases} [\mathbf{e}_b; \mathbf{e}_k] & \text{if token } j \text{ is base } b \in \{A, C, G, T\} \text{ at position } k \\ [\mathbf{e}_|; \mathbf{0}] & \text{if token } j \text{ is separator } | \\ [\mathbf{e}_:; \mathbf{0}] & \text{if token } j \text{ is colon } : \end{cases}$$

where $\mathbf{e}_v$ denotes the standard basis vector with 1 in position $v$ and 0 elsewhere.

Let the first layer be a single-head self-attention layer with weight matrices $\mathbf{W}_Q, \mathbf{W}_K, \mathbf{W}_V \in \mathbb{R}^{L \times d}$ defined as follows. The key matrix $\mathbf{W}_K$ extracts position information:

$$\mathbf{W}_K = \begin{bmatrix} \mathbf{0}_{L \times |\mathcal{V}|} & \mathbf{I}_{L \times L} \end{bmatrix}$$

where $\mathbf{I}$ is the identity matrix. Thus $\mathbf{k}_j = \mathbf{W}_K \mathbf{h}_j$ extracts the position encoding from token $j$. The query matrix $\mathbf{W}_Q$ implements position lookup with shift:

$$\mathbf{W}_Q = \begin{bmatrix} \mathbf{w}_: & \mathbf{0}_{1 \times L} \\ \mathbf{0}_{(L-1) \times |\mathcal{V}|} & \mathbf{S}_{(L-1) \times L} \end{bmatrix}$$

where $\mathbf{w}_: = \mathbf{e}_6 \in \mathbb{R}^{|\mathcal{V}|}$ detects the colon token (position 6 in vocabulary), and $\mathbf{S}$ is the shift matrix with ones on the subdiagonal, such that

$$\mathbf{q}_n = \mathbf{W}_Q \mathbf{h}_n = \begin{cases} \mathbf{e}_1 & \text{if } \mathbf{h}_n \text{ is the colon token} \\ \mathbf{e}_{i+1} & \text{if } \mathbf{h}_n \text{ has position encoding } \mathbf{e}_i \end{cases}$$

The attention scores between the query (from the current position) and each token $j$ in the input for predicting the next token are:

$$s_j = \mathbf{q}^T \mathbf{k}_j = \begin{cases} 1 & \text{if token } j \text{ is at the target position in some trace} \\ 0 & \text{otherwise} \end{cases}$$

where $j \in \{1, \dots, n\}$ indexes all tokens in the current input sequence ($\mathbf{q}$ and $\mathbf{k}_j$ are both one-hot vectors in $\mathbb{R}^L$, so their dot product is 1 if they encode the same position and 0 otherwise).

The self-attention mechanism computes attention weights via softmax. By scaling the scores with a sufficiently large constant $M$, we can construct:

$$\alpha_j = \frac{\exp(M \cdot s_j)}{\sum_{j'=1}^n \exp(M \cdot s_{j'})} \approx \begin{cases} \frac{1}{N_i} & \text{if } s_j = 1 \\ 0 & \text{if } s_j = 0 \end{cases}$$

where $N_i$ is the number of tokens at position $i$ across all traces. This gives uniform weight to all tokens at the target position.

We define the value matrix $\mathbf{W}_V$ as

$$\mathbf{W}_V = \begin{bmatrix} \mathbf{I}_{4 \times 4} & \mathbf{0}_{4 \times (|\mathcal{V}|-4+L)} \\ \mathbf{0}_{(L-4) \times 4} & \mathbf{0}_{(L-4) \times (|\mathcal{V}|-4+L)} \end{bmatrix}$$

such that $\mathbf{v}_j = \mathbf{W}_V \mathbf{h}_j$ extracts the nucleotide one-hot encoding if token $j$ is a base or returns zeros otherwise, padded to dimension $L$. The final attention output is:

$$\mathbf{z} = \sum_j \alpha_j \mathbf{v}_j = \left[ \frac{n_A}{N_i}, \frac{n_C}{N_i}, \frac{n_G}{N_i}, \frac{n_T}{N_i}, 0, \dots, 0 \right] \tag{17}$$

where $n_b$ counts how many traces have nucleotide $b$ at position $i$. This gives us the proportion of votes for each nucleotide, which is what we need to determine the majority vote.

The second layer implements majority selection. The first sublayer uses large weights to threshold the vote proportions in Equation 17:

$$\mathbf{W}_1 = M \cdot \mathbf{I}_{L \times L}, \quad \mathbf{b}_1 = -M(1/4 - \epsilon) \cdot \mathbf{1}_4$$

where $M$ is large and $\epsilon > 0$. After ReLU, only nucleotides with vote proportion $> 1/4 - \epsilon$ become non-zero:

$$[\mathbf{h}^{(1)}]_b = \mathrm{ReLU}(M \cdot [(n_b/N_i) - (1/4 - \epsilon)])$$

The second sublayer with $\mathbf{W}_2 = \begin{bmatrix} \mathbf{I}_{4 \times 4} & \mathbf{0}_{4 \times (L-4)} \end{bmatrix}$ maps to the 4 nucleotides and softmax gives a distribution concentrated on the most frequent nucleotide(s).

### M.4 Proof of Theorem 2

The difference to the Bayes-optimal loss can be written as

$$\mathcal{L}(\theta) - \mathcal{L}^* = \mathbb{E}_{x_i, \{y_i^j\}} \left[ \log \mathrm{P}_{\mathrm{maj}} \left[ x_i \mid \{y_i^j\} \right] - \log \mathrm{P}_\theta \left[ x_i \mid \{y_i^j\} \right] \right]$$

$$= \mathbb{E}_{\{y_i^j\}} \left[ \sum_{x_i} \mathrm{P}_{\mathrm{true}} \left[ x_i \mid \{y_i^j\} \right] \log \frac{\mathrm{P}_{\mathrm{maj}} \left[ x_i \mid \{y_i^j\} \right]}{\mathrm{P}_\theta \left[ x_i \mid \{y_i^j\} \right]} \right]$$

$$= \mathbb{E}_{\{y_i^j\}} \left[ \mathrm{KL} \left[ \mathrm{P}_{\mathrm{maj}} \left[ \cdot \mid \{y_i^j\} \right] \ \| \ \mathrm{P}_\theta \left[ \cdot \mid \{y_i^j\} \right] \right] \right],$$

where the last equality follows from optimality of majority voting.

By Pinsker's inequality (see, e.g., Lemma 2.5 in Tsybakov (2009)) we have

$$\left\| \mathrm{P}_{\mathrm{maj}} \left[ \cdot \mid \{y_i^j\} \right] - \mathrm{P}_\theta \left[ \cdot \mid \{y_i^j\} \right] \right\|_{TV} \leq \sqrt{\tfrac{1}{2} \mathrm{KL} \left[ \mathrm{P}_{\mathrm{maj}} \left[ \cdot \mid \{y_i^j\} \right] \ \| \ \mathrm{P}_\theta \left[ \cdot \mid \{y_i^j\} \right] \right]}.$$

Taking expectations and applying Jensen's inequality (since $\sqrt{\cdot}$ is concave) we get

$$\mathbb{E}_{\{y_i^j\}} \left[ \left\| \mathrm{P}_{\mathrm{maj}} \left[ \cdot \mid \{y_i^j\} \right] - \mathrm{P}_\theta \left[ \cdot \mid \{y_i^j\} \right] \right\|_{TV} \right]$$

$$\leq \mathbb{E}_{\{y_i^j\}} \left[ \sqrt{\tfrac{1}{2} (\mathcal{L}(\theta) - \mathcal{L}^*)} \right]$$

$$\leq \sqrt{\tfrac{\delta}{2}}.$$

### M.5 Empirical validation

To validate our theoretical results from Section 6 empirically, we train three models with compute budgets of $6 \times 10^{17}$, $1 \times 10^{18}$, and $3 \times 10^{18}$ FLOPs on data with substitution errors only (rates sampled uniformly from $[0.01, 0.1]$). We use the same architecture as in Section 5.2 but with batch size 16 and learning rate $10^{-4}$, and evaluate on 50K test examples generated with the same error distribution as the training data.

The excess losses $\mathcal{L}(\theta) - \mathcal{L}^*$ relative to the Bayes-optimal loss decrease with compute budget: $1.8 \times 10^{-4}$ ($6 \times 10^{17}$ FLOPs), $1.6 \times 10^{-4}$ ($1 \times 10^{18}$ FLOPs), and $1.5 \times 10^{-4}$ ($3 \times 10^{18}$ FLOPs). By Theorem 2, these

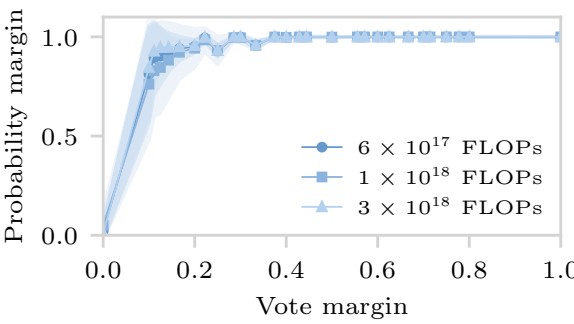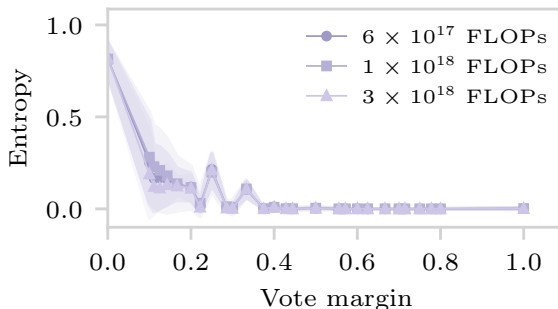

Figure 17: Vote margin versus model confidence metrics for substitution-only reconstruction.

small excess losses imply that all three models approximate majority voting behavior. To validate this empirically, we compare the vote margin in the data with the model's probability margins and entropies at each token position. The vote margin is defined as the difference between the most frequent and second most frequent nucleotides at a position, normalized by the cluster size. For the model, the probability margin is the difference between the highest and second highest predicted nucleotide probabilities, and the entropy is computed only over the four nucleotides, excluding other vocabulary tokens.

Figure 17 shows vote margin versus model confidence aggregated across all cluster sizes. As vote margins increase, the model shows higher probability margins and lower entropy (less uncertainty), matching the expected behavior of majority voting. For small clusters (N=2,3), we observe strong linear correlations (Pearson r=0.82 for the model trained with compute $3 \times 10^{18}$ FLOPs) between vote margins and confidence metrics, with the correlation decreasing for larger clusters (r=0.29 for N=4-6, r=0.08 for N=7-10 for compute $3 \times 10^{18}$ FLOPs) because both vote and probability margins concentrate near 1.0, reducing variance.

For the model trained with compute $3 \times 10^{18}$ FLOPs, we further probe the last-layer hidden representations with a linear classifier to test whether the model encodes per-position base counts directly. Training a linear probe to predict base frequencies achieves a test mean squared error of 0.0034 (KL divergence of 0.105), consistent with our theoretical construction in Section M.3 where the attention output directly encodes base frequency information.

Our empirical findings support our theoretical analysis that, in the substitution-only setting, transformers learn the optimal majority voting strategy. These initial results suggest that language models trained with next-token prediction can learn optimal algorithmic strategies for sequence reconstruction tasks, though our theoretical analysis is limited to substitution errors.

## N  Detailed numerical results

For better readability and comparison, we provide tables with the numerical results (Levenshtein distances and failure rates) for the experiments in the main paper. We include tables for the evaluation on synthetic data for $L = 110$ (Figure 2) and for the real-world data experiments with the Noisy-DNA dataset (Figure 4) and the Microsoft dataset (Figure 5). The tables report results for our pretrained and, for the real-world datasets, fine-tuned models, alongside baselines. Reported standard deviations are across test examples (not random seeds). For TReconLM, the main paper plots averages over three runs with different seeds, whereas the tables show results with seed 100.

Table 5: Results for synthetic data of length $L = 110$ (see Figure 2).

| | | | | Levenshtein distance $d_\mathrm{L}$ | | | | |
|---|---|---|---|---|---|---|---|---|
| $N$ | RobuSeqNet | VS | MUSCLE | BMALA | TrellisBMA | ITR | DNAformer | **TReconLM** |
| 2 | 3.96e-1 (7.08e-2) | 1.59e-1 (5.27e-2) | 2.07e-1 (6.62e-2) | 2.39e-1 (7.67e-2) | 4.19e-1 (6.63e-2) | 1.55e-1 (5.41e-2) | 2.66e-1 (8.58e-2) | **1.42e-1 (5.31e-2)** |
| 3 | 3.85e-1 (7.23e-2) | 1.58e-1 (5.28e-2) | 1.50e-1 (6.21e-2) | 1.74e-1 (7.77e-2) | 4.04e-1 (7.19e-2) | 2.47e-1 (8.86e-2) | 1.45e-1 (9.23e-2) | **6.56e-2 (4.11e-2)** |
| 4 | 3.73e-1 (7.41e-2) | 1.69e-1 (5.68e-2) | 1.07e-1 (5.19e-2) | 1.46e-1 (7.65e-2) | 1.43e-1 (8.33e-2) | 7.36e-2 (4.86e-2) | 8.70e-2 (7.63e-2) | **3.82e-2 (3.12e-2)** |
| 5 | 3.62e-1 (7.57e-2) | 1.62e-1 (5.49e-2) | 8.11e-2 (4.66e-2) | 1.19e-1 (7.38e-2) | 1.01e-1 (7.01e-2) | 4.45e-2 (3.97e-2) | 4.95e-2 (5.66e-2) | **2.17e-2 (2.33e-2)** |
| 6 | 3.55e-1 (7.59e-2) | 1.62e-1 (5.55e-2) | 7.23e-2 (4.32e-2) | 9.84e-2 (6.83e-2) | 1.10e-1 (8.20e-2) | 2.28e-2 (2.62e-2) | 2.90e-2 (4.09e-2) | **1.26e-2 (1.73e-2)** |
| 7 | 3.49e-1 (7.62e-2) | 1.65e-1 (5.68e-2) | 6.64e-2 (4.07e-2) | 8.22e-2 (6.37e-2) | 8.11e-2 (6.84e-2) | 1.44e-2 (2.01e-2) | 1.83e-2 (3.00e-2) | **7.77e-3 (1.32e-2)** |
| 8 | 3.45e-1 (7.45e-2) | 1.70e-1 (5.85e-2) | 5.81e-2 (3.79e-2) | 7.36e-2 (6.05e-2) | 6.30e-2 (5.93e-2) | 9.19e-3 (1.51e-2) | 1.18e-2 (2.27e-2) | **4.79e-3 (1.01e-2)** |
| 9 | 3.39e-1 (7.43e-2) | 1.75e-1 (6.14e-2) | 4.98e-2 (3.48e-2) | 6.23e-2 (5.62e-2) | 4.86e-2 (5.15e-2) | 5.53e-3 (1.11e-2) | 7.46e-3 (1.69e-2) | **2.90e-3 (7.63e-3)** |
| 10 | 3.34e-1 (7.50e-2) | 1.78e-1 (6.15e-2) | 4.49e-2 (3.26e-2) | 5.37e-2 (5.27e-2) | 3.62e-2 (4.25e-2) | 3.77e-3 (8.86e-3) | 5.05e-3 (1.41e-2) | **1.75e-3 (5.87e-3)** |
| | | | | Failure rate | | | | |
| $N$ | RobuSeqNet | VS | MUSCLE | BMALA | TrellisBMA | ITR | DNAformer | **TReconLM** |
| 2 | 1.00e+0 | 1.00e+0 | 1.00e+0 | 1.00e+0 | 1.00e+0 | 1.00e+0 | 1.00e+0 | **9.99e-1** |
| 3 | 1.00e+0 | 1.00e+0 | 9.99e-1 | 9.96e-1 | 1.00e+0 | 9.99e-1 | 9.73e-1 | **9.62e-1** |
| 4 | 1.00e+0 | 1.00e+0 | 9.94e-1 | 9.85e-1 | 9.88e-1 | 9.45e-1 | 8.97e-1 | **8.44e-1** |
| 5 | 1.00e+0 | 1.00e+0 | 9.79e-1 | 9.63e-1 | 9.62e-1 | 8.45e-1 | 7.67e-1 | **6.72e-1** |
| 6 | 1.00e+0 | 1.00e+0 | 9.71e-1 | 9.25e-1 | 9.50e-1 | 6.66e-1 | 6.22e-1 | **4.91e-1** |
| 7 | 1.00e+0 | 1.00e+0 | 9.64e-1 | 8.90e-1 | 9.01e-1 | 5.20e-1 | 4.80e-1 | **3.42e-1** |
| 8 | 1.00e+0 | 1.00e+0 | 9.44e-1 | 8.58e-1 | 8.58e-1 | 3.98e-1 | 3.83e-1 | **2.34e-1** |
| 9 | 1.00e+0 | 9.99e-1 | 9.17e-1 | 8.10e-1 | 7.85e-1 | 2.80e-1 | 2.61e-1 | **1.50e-1** |
| 10 | 1.00e+0 | 9.99e-1 | 9.00e-1 | 7.65e-1 | 7.06e-1 | 2.12e-1 | 1.88e-1 | **9.54e-2** |

Table 6: Results for the Noisy-DNA dataset (see Figure 4). Pretrained models (p) and fine-tuned models (f).

| | | | | Levenshtein distance $d_\mathrm{L}$ | | | | | |
|---|---|---|---|---|---|---|---|---|---|
| $N$ | RobuSeqNet (f) | VS | MUSCLE | BMALA | TrellisBMA | ITR | TReconLM (p) | DNAformer (f) | **TReconLM (f)** |
| 2 | 4.05e-1 (1.27e-1) | 3.92e-1 (1.75e-1) | 4.35e-1 (1.32e-1) | 4.39e-1 (1.15e-1) | 5.44e-1 (1.06e-1) | 3.83e-1 (1.77e-1) | 3.89e-1 (1.39e-1) | 4.07e-1 (1.64e-1) | **3.30e-1 (1.95e-1)** |
| 3 | 3.77e-1 (1.34e-1) | 3.86e-1 (1.71e-1) | 4.12e-1 (1.45e-1) | 4.07e-1 (1.35e-1) | 5.23e-1 (1.11e-1) | 5.57e-1 (1.50e-1) | 3.45e-1 (1.52e-1) | 3.47e-1 (1.72e-1) | **2.94e-1 (2.07e-1)** |
| 4 | 3.68e-1 (1.30e-1) | 3.85e-1 (1.64e-1) | 3.71e-1 (1.40e-1) | 3.98e-1 (1.36e-1) | 4.51e-1 (2.00e-1) | 3.71e-1 (1.60e-1) | 3.23e-1 (1.55e-1) | 3.19e-1 (1.74e-1) | **2.70e-1 (2.11e-1)** |
| 5 | 3.48e-1 (1.33e-1) | 3.88e-1 (1.62e-1) | 3.48e-1 (1.44e-1) | 3.78e-1 (1.45e-1) | 3.90e-1 (2.08e-1) | 3.58e-1 (1.74e-1) | 2.98e-1 (1.56e-1) | 2.79e-1 (1.83e-1) | **2.33e-1 (1.16e-1)** |
| 6 | 3.45e-1 (1.33e-1) | 3.78e-1 (1.65e-1) | 3.47e-1 (1.41e-1) | 3.72e-1 (1.53e-1) | 4.27e-1 (2.09e-1) | 3.12e-1 (1.70e-1) | 2.89e-1 (1.52e-1) | 2.62e-1 (1.80e-1) | **2.16e-1 (2.15e-1)** |
| 7 | 3.26e-1 (1.33e-1) | 3.66e-1 (1.67e-1) | 3.34e-1 (1.45e-1) | 3.54e-1 (1.61e-1) | 3.74e-1 (2.20e-1) | 2.93e-1 (1.77e-1) | 2.71e-1 (1.55e-1) | 2.38e-1 (1.86e-1) | **1.91e-1 (2.12e-1)** |
| 8 | 3.23e-1 (1.32e-1) | 3.77e-1 (1.60e-1) | 3.30e-1 (1.46e-1) | 3.51e-1 (1.56e-1) | 3.09e-1 (1.87e-1) | 2.76e-1 (1.75e-1) | 2.64e-1 (1.54e-1) | 2.27e-1 (1.85e-1) | **1.76e-1 (2.12e-1)** |
| 9 | 3.18e-1 (1.29e-1) | 3.77e-1 (1.56e-1) | 3.19e-1 (1.43e-1) | 3.45e-1 (1.68e-1) | 2.91e-1 (1.85e-1) | 2.69e-1 (1.76e-1) | 2.59e-1 (1.55e-1) | 2.14e-1 (1.82e-1) | **1.66e-1 (2.09e-1)** |
| 10 | 3.22e-1 (1.30e-1) | 3.80e-1 (1.51e-1) | 3.21e-1 (1.47e-1) | 3.46e-1 (1.68e-1) | 3.37e-1 (2.17e-1) | 2.67e-1 (1.78e-1) | 2.61e-1 (1.58e-1) | 2.18e-1 (2.29e-1) | **1.62e-1 (2.10e-1)** |
| | | | | Failure rate | | | | | |
| $N$ | RobuSeqNet (f) | VS | MUSCLE | BMALA | TrellisBMA | ITR | TReconLM (p) | DNAformer (f) | **TReconLM (f)** |
| 2 | 9.99e-1 | 9.94e-1 | 1.00e+0 | 1.00e+0 | 1.00e+0 | 9.93e-1 | 1.00e+0 | 9.99e-1 | **9.83e-1** |
| 3 | 9.99e-1 | 9.96e-1 | 9.99e-1 | 1.00e+0 | 1.00e+0 | 1.00e+0 | 1.00e+0 | 9.92e-1 | **9.47e-1** |
| 4 | 1.00e+0 | 9.97e-1 | 9.98e-1 | 1.00e+0 | 9.98e-1 | 9.91e-1 | 9.99e-1 | 9.82e-1 | **9.19e-1** |
| 5 | 9.98e-1 | 9.95e-1 | 9.94e-1 | 1.00e+0 | 9.95e-1 | 9.89e-1 | 1.00e+0 | 9.61e-1 | **8.60e-1** |
| 6 | 9.98e-1 | 9.95e-1 | 9.96e-1 | 9.99e-1 | 9.92e-1 | 9.79e-1 | 9.98e-1 | 9.51e-1 | **8.15e-1** |
| 7 | 9.98e-1 | 9.93e-1 | 9.97e-1 | 1.00e+0 | 9.91e-1 | 9.70e-1 | 1.00e+0 | 9.13e-1 | **7.61e-1** |
| 8 | 9.98e-1 | 9.96e-1 | 9.94e-1 | 1.00e+0 | 9.87e-1 | 9.66e-1 | 9.99e-1 | 8.91e-1 | **7.05e-1** |
| 9 | 9.97e-1 | 9.97e-1 | 9.93e-1 | 9.99e-1 | 9.91e-1 | 9.61e-1 | 9.99e-1 | 8.84e-1 | **6.81e-1** |
| 10 | 9.97e-1 | 9.95e-1 | 9.91e-1 | 1.00e+0 | 9.86e-1 | 9.67e-1 | 9.99e-1 | 8.77e-1 | **6.48e-1** |

Table 7: Results for the Microsoft dataset (see Figure 5). Pretrained models (p) and fine-tuned models (f).

| | | | | Levenshtein distance $d_{\mathrm{L}}$ | | | | | |
|---|---|---|---|---|---|---|---|---|---|
| $N$ | RobuSeqNet (f) | VS | MUSCLE | BMALA | TrellisBMA | ITR | TReconLM (p) | DNAformer (f) | **TReconLM (f)** |
| 2 | 1.92e-1 (8.01e-2) | 5.67e-2 (2.90e-2) | 6.69e-2 (3.11e-2) | 1.25e-1 (6.62e-2) | 1.91e-1 (8.78e-2) | 5.39e-2 (3.03e-2) | 5.34e-2 (2.87e-2) | 7.34e-2 (4.77e-2) | **4.19e-2 (2.88e-2)** |
| 3 | 1.37e-1 (6.68e-2) | 5.73e-2 (2.96e-2) | 4.82e-2 (2.39e-2) | 5.19e-2 (4.01e-2) | 1.55e-1 (7.25e-2) | 9.04e-2 (3.97e-2) | 1.55e-2 (1.52e-2) | 1.82e-2 (2.26e-2) | **1.30e-2 (1.60e-2)** |
| 4 | 1.14e-1 (6.39e-2) | 7.33e-2 (5.91e-2) | 1.45e-2 (1.30e-2) | 3.98e-2 (3.57e-2) | 1.49e-2 (1.60e-2) | 7.84e-3 (1.12e-2) | 7.34e-3 (1.08e-2) | 8.61e-3 (1.84e-2) | **4.50e-3 (9.10e-3)** |
| 5 | 1.04e-1 (5.92e-2) | 6.30e-2 (3.95e-2) | 9.22e-3 (1.03e-2) | 2.81e-2 (3.22e-2) | 1.11e-2 (1.37e-2) | 5.53e-3 (9.27e-3) | 4.50e-3 (8.98e-3) | 5.25e-3 (1.54e-2) | **2.50e-3 (7.58e-3)** |
| 6 | 8.85e-2 (5.64e-2) | 6.10e-2 (3.92e-2) | 8.20e-3 (9.74e-3) | 2.18e-2 (2.56e-2) | 7.05e-3 (1.15e-2) | 2.83e-3 (5.96e-3) | 2.29e-3 (6.00e-3) | 2.26e-3 (8.31e-3) | **1.50e-3 (5.00e-3)** |
| 7 | 8.04e-2 (5.41e-2) | 6.17e-2 (4.40e-2) | 8.61e-3 (1.00e-2) | 1.54e-2 (2.14e-2) | 5.48e-3 (1.01e-2) | 1.96e-3 (5.14e-3) | 1.74e-3 (5.38e-3) | 1.39e-3 (4.78e-3) | **9.00e-4 (4.37e-3)** |
| 8 | 7.39e-2 (5.24e-2) | 6.26e-2 (4.62e-2) | 4.78e-3 (7.10e-3) | 1.31e-2 (2.15e-2) | 4.82e-3 (1.01e-2) | 1.68e-3 (4.72e-3) | 1.50e-3 (5.05e-3) | 1.30e-3 (8.58e-3) | **7.00e-4 (3.05e-3)** |
| 9 | 6.57e-2 (4.69e-2) | 6.50e-2 (4.77e-2) | 3.86e-3 (6.64e-3) | 1.06e-2 (1.71e-2) | 4.49e-3 (9.51e-3) | 1.52e-3 (4.13e-3) | 1.19e-3 (4.62e-3) | 1.05e-3 (5.58e-3) | **6.36e-4 (3.48e-3)** |
| 10 | 7.36e-2 (5.49e-2) | 7.04e-2 (5.07e-2) | 3.32e-3 (5.85e-3) | 8.95e-3 (1.83e-2) | 3.42e-3 (8.34e-3) | 1.27e-3 (3.90e-3) | 1.14e-3 (4.47e-3) | 1.08e-3 (6.23e-3) | **3.00e-4 (2.92e-3)** |

| | | | | Failure rate | | | | | |
|---|---|---|---|---|---|---|---|---|---|
| $N$ | RobuSeqNet (f) | VS | MUSCLE | BMALA | TrellisBMA | ITR | TReconLM (p) | DNAformer (f) | **TReconLM (f)** |
| 2 | 9.95e-1 | 9.75e-1 | 9.92e-1 | 9.95e-1 | 9.99e-1 | 9.51e-1 | 9.61e-1 | 9.70e-1 | **8.91e-1** |
| 3 | 9.99e-1 | 9.99e-1 | 9.82e-1 | 9.03e-1 | 9.93e-1 | 9.91e-1 | 6.35e-1 | 6.14e-1 | **4.94e-1** |
| 4 | 9.81e-1 | 9.69e-1 | 7.39e-1 | 8.61e-1 | 6.09e-1 | 4.22e-1 | 3.89e-1 | 3.41e-1 | **2.32e-1** |
| 5 | 9.75e-1 | 9.71e-1 | 5.76e-1 | 7.30e-1 | 4.96e-1 | 3.38e-1 | 2.41e-1 | 2.10e-1 | **1.35e-1** |
| 6 | 9.53e-1 | 9.73e-1 | 5.32e-1 | 6.36e-1 | 3.54e-1 | 2.19e-1 | 1.49e-1 | 1.17e-1 | **8.81e-2** |
| 7 | 9.49e-1 | 9.55e-1 | 5.78e-1 | 5.00e-1 | 2.79e-1 | 1.49e-1 | 1.04e-1 | 8.96e-2 | **5.30e-2** |
| 8 | 9.32e-1 | 9.60e-1 | 3.65e-1 | 5.17e-1 | 2.31e-1 | 1.36e-1 | 9.23e-2 | 5.71e-2 | **3.52e-2** |
| 9 | 9.14e-1 | 9.50e-1 | 3.05e-1 | 4.15e-1 | 2.35e-1 | 1.35e-1 | 7.00e-2 | 4.97e-2 | **3.84e-2** |
| 10 | 9.15e-1 | 9.57e-1 | 2.79e-1 | 3.22e-1 | 1.71e-1 | 1.10e-1 | 6.85e-2 | 4.34e-2 | **1.37e-2** |

**Example prompt for GPT-4o mini:**

We consider a reconstruction problem of DNA sequences. We want to reconstruct a DNA sequence consisting of 60 characters (either A,C,T or G) from 5 noisy DNA sequences.
These noisy DNA sequences were generated by introducing random errors (insertion, deletion, and substitution of single characters).
The task is to provide an estimate of the ground truth DNA sequence.

Here are some examples:
Example #1
Input DNA sequences:
1. GATACGGATTGTGCTCGAGTGGATACTGGTATAGAGAAGAGAGTAATGCTAAGGTAG
2. ATATAGGACTGTTCCTCGAAGTGGATACTGTACAAAAATCAGAAGCGAGTAAGGTAG
3. GATCAGGATTGTACTCGAGTGCTACTGTACAAAGCGTCAGAGGTGCCATAGGTACG
4. GATAAAGGGACGTTGCCCGAGTGATACTGTCAAAGCGTAAAAGAGATGCTAGGTG
5. GGATCAAGGATTGCTTGTCGAGTGTGATACTGTACAATGATCAGAAGAGATTAATAG
Correct output:
GATAAAGGATTGTTGCTCGAGTGGATACTGTACAAAGAGTCAGAAGAGATGCTAAGGTAG

Example #2
Input DNA sequences:
1. AAACCCTTACGGGTCGAATACATCTTATCCGAGCGCCTCAAGGAGTAGCGATTCCTAC
2. AAACCCATAGGGTCCAAAAATATTTACCGTGCACTCCGAAAGGGAGTATCGTTGATA
3. AAACACTTGGGGTCGAAAAAATACTATCCGTGTACCCCAGAGGTGTAGTGTCTCATAC
4. AACCTGAGGGTCGAAACTGTTGATCCGTGCACCTCATGAGGGTGTCGCGGCATGC
5. AAACCTTAGGGCTCGAATACATATTTACCGTGCACCTCCAGAGGAGTAGCGTTTCAA
Correct output:
AAACCCTTAGGGTCGAATACATATTTATCCGTGCACCTCCAGAGGAGTAGCGTTTCATAC

Example #3
Input DNA sequences:
1. TGCCCCGACGATATGCCGGCGGATACACTCTCACGATCGTCAAGTATATCCGTTAA
2. ATGCCCGACGCTTCTGGCCGGATACACTCAACAATCGTCACCGTTTATCCGATAA
3. ATGCCCGACGAATGCTGGCCGGATACACTTACACGATGTCAATGATATCCGAGTG
4. ATGCCCACGAGTATGCTGCCGGATCCTCACAAATCGTCAAGTTATATCCCGATAT
5. ATGCCCGATAATATATGGCGGACTCCACTCTACACGTCGTCAAGTTATATCCCGTTAG
Correct output:
ATGCCCGACGATATGCTGGCCGGATACACTCTACACGATCGTCAAGTTATATCCCGTTAT

Task:
Reconstruct the DNA sequence from the following noisy input sequences.
Input DNA sequences:
1. GGTCCCTAGAAGGATTGGATGCTGTTCGCGGGTATCTAATGTTGTGCCTTGGTGCAT
2. AGGTCGCCCAGAAGTGATATGGTCGCTGGCGCGGCATCTAATTTGTGACATCTTGAT
3. AGGTTACCCTGATAGTGATGTAGTGTGCATTTCGCGGCTCTATGTTGTGCCTGTTGCT
4. AGGTCCTAGTAAGGTATATGCATGCGGTCGCGGCTCTAATGTTGTGCTTGAGTTGCT
5. AGCTCCGTAGAGGAATGATGCTGTTCGCCGGCATTAGATGTGTGCCTCGGTTGCT
Provide an estimate of the ground truth DNA sequence consisting of 60 characters in the
format ***estimated DNA sequence*** - use three * on each side of the estimated DNA sequence.

Figure 18: Three-shot prompt example for GPT-4o mini.

