# OpenReview forum: "Trace Reconstruction with Language Models"
_TMLR — Accepted by TMLR_

### Review · Reviewer_EQ1k · 2026-03-29

**Summary Of Contributions:**

This paper proposes a method for sequence reconstruction for DNA sequences, where the task is to reconstruct a ground truth DNA sequence given multiple perturbed observations of the original sequence. The methodology is pretty simple, i.e. to train a transformer model, to reconstruct the original sequence given the observations. The model is pretrained with randomly synthesized data, with randomly synthesized perturbations. In addition, to adapt to real-world noises, the authors propose fine-tuning on real-world data based on the pre-trained model. The authors compare their proposed method with multiple baselines, including traditional algorithms, deep-learning based methods, and state-of-the-art LLMs where the proposed method shows promising performance. In addition, scaling law experiments show that the proposed method does not scale indefinitely with respect to number of parameters --- an interesting observation.

**Additional Comments:**

NA

**Audience:**

Yes

**Audience Explanation:**

Machine learning for biological science is a popular topic for those in the intersection of the two areas. This paper additionally studies the combination of the topic with pre-training of language models, further increasing the audience of this paper.

**Broader Impact Concerns:**

Not really. The experiments are all done with synthetic data and public datasets.

**Claims And Evidence:**

Yes

**Claims Explanation:**

I am not an expert in this area of biomedical machine learning, and thus cannot verify the correctness of some assumptions made in this paper, such as "most DNA sequences are uniform at random", etc.

From the machine learning point-of-view, the paper is conceptually simple and mostly sound. I appreciate the authors including comparisons with a wide range of baseline methods, such as traditional algorithms, deep-learning based algorithms, as well as LLM-based prompting methods. I also appreciate the inclusion of scaling behaviors.

From the perspective of machine learning, it would be good if the authors include the following analyses and discussions.

1. The validity of some of the assumptions. For example, the "uniformity at random" of DNA sequences in practice, and the "uniformity at random" of perturbations.
2. Generalization. For example, after fine-tuning to one particular distribution, how does it perform on another distribution?
3. Robustness to the order of input sequences at inference time.
4. Robustness to the number of input sequences. For example, if the model is trained with 10 input sequences, what happens if I only input 4? What happens if I input 40?

**Requested Changes:**

Ref. "Explanation"

---

> ### Author Response · Authors · 2026-04-20
>
> We thank the reviewer for their careful reading of our paper and the detailed comments. We address each of the four questions below.
>
> 1. **Regarding uniformity of DNA sequences:** The assumption that DNA sequences are approximately uniform at random is justified by a common design choice in DNA data storage systems, where the binary input data is XORed with a pseudorandom sequence before encoding it into nucleotides (see also Section 3 in the main paper). This randomization step is applied to, e.g., break up repetitive patterns that are more error-prone [1, 2]. In addition, if the input data is compressed, it already appears approximately uniform at random [3]. To also verify this empirically, we compute the per-sequence nucleotide frequencies across all ground-truth sequences in the real-world datasets we consider, and report the mean and standard deviation:
>
>    | Dataset        | A                  | C                  | G                  | T                  |
>    |----------------|--------------------|--------------------|--------------------|--------------------|
>    | Noisy-DNA      | 28.07\% (5.36\%)   | 26.50\% (5.35\%)   | 22.23\% (5.33\%)   | 23.20\% (5.23\%)   |
>    | Microsoft      | 24.65\% (4.14\%)   | 25.12\% (4.13\%)   | 24.98\% (4.13\%)   | 25.25\% (4.13\%)   |
>    | Chandak et al. | 24.26\% (3.82\%)   | 24.98\% (3.78\%)   | 25.23\% (3.82\%)   | 25.54\% (3.99\%)   |
>
>    As shown in the table, all mean frequencies are close to 25% with small standard deviations, consistent with the assumption we use for synthetic data generation.
>
>    **Regarding the uniformity of perturbations:** While error patterns may depend on the specific protocol and technology used, the assumption of uniformly random perturbations holds approximately, specifically for substitution errors [4]. A uniform error model is also technology-agnostic, avoiding overfitting to the error distribution of a specific technology, and is therefore well suited for pretraining a foundation model. As shown in Appendix J.2, TReconLM pretrained with a uniform error model already achieves a lower failure rate (0.111) than DNAformer (0.146), which was pretrained on synthetic data generated using error statistics from the real Microsoft dataset, suggesting that a uniform error model is also an effective strategy for pretraining.
>
> 2. **Generalization:** We ran additional cross-dataset experiments, where we evaluated models fine-tuned on one real-world dataset against the others. Since models are trained on fixed sequence lengths, cross-dataset evaluation can be confounded by ground-truth length mismatches. We therefore report results both raw and length-controlled, where for the length-controlled results we align each input trace to the ground-truth sequence via minimum edit distance and crop both the traces and the ground-truth sequence to the model's training length before inference.
>
>    For the Chandak et al. and Microsoft datasets, which share the same sequencing technology (Oxford Nanopore) but differ in synthesis technology (Twist Bioscience vs. Custom Array), cross-fine-tuned models perform similarly to the pretrained model after controlling for length, but we do not observe cross-dataset generalization from fine-tuning. For Noisy-DNA and Microsoft, which differ in both synthesis (photolithographic vs. Twist Bioscience) and sequencing technology (Illumina vs. Oxford Nanopore), the pretrained 60nt model outperforms the Noisy-DNA cross-fine-tuned model on the Microsoft test data cropped to length 60nt (25.2% vs. 60.6% failure rate), indicating that the Noisy-DNA error distribution is more out-of-distribution relative to Microsoft data than the synthetic pretraining distribution is. This is consistent with the results in the main paper: the pretrained model already performs competitively on the Microsoft dataset without fine-tuning (Figure 5), whereas on the Noisy-DNA dataset it does not generalize as well (Figure 4).
>
>    **Microsoft dataset**
>
>    | Model                                | Mean Lev | Failure % |
>    |--------------------------------------|----------|-----------|
>    | Finetuned Microsoft (in-domain)      | 0.0107   | 29.6      |
>    | Pretrained 110nt                     | 0.0141   | 38.1      |
>    | Variable length                      | 0.0229   | 48.8      |
>    | Finetuned Chandak et al. (cross)     | 0.0686   | 91.1      |
>    | Finetuned Noisy DNA (cross)          | 0.2727   | 100.0     |
>    | length-controlled to 60nt            |          |           |
>    | Pretrained 60nt                      | 0.0120   | 25.2      |
>    | Variable length                      | 0.0235   | 36.4      |
>    | Finetuned Noisy DNA (cross)          | 0.0306   | 60.6      |

---

> > ### Author Response · Authors · 2026-04-20
> >
> > **Noisy-DNA dataset:**
> >
> >    | Model                                | Mean Lev | Failure % |
> >    |--------------------------------------|----------|-----------|
> >    | Finetuned Noisy DNA (in-domain)      | 0.2353   | 83.0      |
> >    | Variable length                      | 0.2963   | 97.2      |
> >    | Pretrained 60nt                      | 0.3421   | 100.0     |
> >    | Finetuned Chandak et al. (cross)     | 0.4628   | 99.9      |
> >    | Finetuned Microsoft (cross)          | 0.4780   | 99.9      |
> >
> >    **Chandak et al. dataset:**
> >
> >    | Model                                | Mean Lev | Failure % |
> >    |--------------------------------------|----------|-----------|
> >    | Finetuned Chandak et al. (in-domain) | 0.0687   | 66.0      |
> >    | Variable length                      | 0.0912   | 90.6      |
> >    | Finetuned Microsoft (cross)          | 0.1365   | 100.0     |
> >    | Pretrained 110nt                     | 0.1503   | 100.0     |
> >    | length-controlled to 110nt           |          |           |
> >    | Finetuned Chandak et al. (in-domain) | 0.0735   | 68.0      |
> >    | Pretrained 110nt                     | 0.0755   | 82.7      |
> >    | Finetuned Microsoft (cross)          | 0.0788   | 83.9      |
> >    | Variable length                      | 0.0931   | 89.5      |
> >    | length-controlled to 60nt            |          |           |
> >    | Pretrained 60nt                      | 0.0713   | 67.6      |
> >    | Variable length                      | 0.0977   | 79.9      |
> >    | Finetuned Noisy DNA (cross)          | 0.1204   | 94.0      |
> >    | Finetuned Chandak et al. (in-domain) | 0.2912   | 94.5      |
> >
> >
> >
> > 3. **Regarding robustness to the order of input sequences at inference time:** We address this in Appendix B, where we evaluate TReconLM under maximum 10 random permutations of the input traces. Reconstructions from different permutations differ by an average of 7.2 nucleotides out of 110 (pairwise Hamming distance), with a per-position vote agreement of 0.96 across permutations (i.e., at each position, 96% of permutations agree on the same nucleotide), demonstrating that the model is largely robust to input ordering. Any remaining dependence on input ordering can be further mitigated by permutation-based majority voting, where we decode multiple random permutations of the input traces and select the most frequent nucleotide at each position, improving the failure rate by 0.5% at the cost of a small increase in Levenshtein distance of 0.03% (Figure 7, Appendix B).
> >
> >    To further quantify robustness to input ordering, we have added a new experiment to Appendix B, where we evaluate TReconLM 20 times on the test set, each time with a different random permutation of the input traces for every example, and report the mean and standard deviation across runs. The overall failure rate is $52.89\% \pm 0.52\%$ and the overall Levenshtein distance is $0.0339 \pm 0.0003$, with the small standard deviations indicating that TReconLM is robust to the ordering of input traces.

---

> > > ### Author Response · Authors · 2026-04-20
> > >
> > > 4. **Regarding robustness to the number of input sequences:** Our main model is trained on varying cluster sizes $N \in [2,10]$ and thus handles variable numbers of input sequences at inference time, as evaluated in Figure 2. To directly address the reviewer's question of what happens when a model receives a different number of input sequences than seen during training, we evaluate a model trained on fixed $N=2$ on larger cluster sizes $N \in [3,10]$ (i.e., more inputs than seen during training) and a model trained on fixed $N=10$ on smaller cluster sizes $N \in [2,9]$ (i.e., fewer inputs than seen during training).
> > >
> > >    When the fixed $N=2$ model is evaluated on larger cluster sizes, performance collapses. This is because the additional traces (traces 3 and beyond) correspond to token positions that were always padding tokens during training and therefore never received meaningful gradient updates, so their positional embeddings are effectively zero. As a result, the model cannot distinguish between the same nucleotide at different positions, i.e., all A's in these additional traces look identical to each other, which we verify by computing pairwise cosine similarity between same-nucleotide tokens in these traces across all layers (cosine similarity $\approx 1.0$ at the embedding layer and at $\approx 0.91$ in the final layer, compared to $\approx 0.49$ and $\approx 0.39$ respectively for traces within the training range). In addition, since the model was only trained on $N=2$ traces, it never learned to ignore additional traces with meaningless token representations, and therefore still attends to them, resulting in corrupted reconstruction.
> > >
> > >    When the fixed $N=10$ model is evaluated on smaller cluster sizes, the model is effectively trained on an easier problem (larger cluster size) and evaluated on a harder one (smaller cluster sizes), which is reflected in a performance loss for out-of-distribution cluster sizes $N < 10$, and a small performance gain at $N=10$ compared to the variable cluster size model. We summarize the differences in Levenshtein distances $d_L$ and failure rates between the fixed $N=10$ model and the variable cluster size model below:
> > >
> > >    |                       | $N=2$   | $N=3$   | $N=4$   | $N=5$   | $N=6$   | $N=7$   | $N=8$   | $N=9$   | $N=10$  |
> > >    |-----------------------|---------|---------|---------|---------|---------|---------|---------|---------|---------|
> > >    | $\Delta d_L$          | +0.4527 | +0.4871 | +0.4423 | +0.2574 | +0.0350 | +0.0077 | +0.0065 | +0.0159 | −0.0003 |
> > >    | $\Delta$ Failure rate | +0.07%  | +3.81%  | +15.69% | +32.54% | +31.60% | +24.62% | +26.21% | +52.33% | −0.99%  |
> > >
> > >
> > >
> > > We hope this addresses the reviewer’s concern; we would be very happy to clarify further.
> > >
> > > [1] Antkowiak, Philipp L., et al. "Low cost DNA data storage using photolithographic synthesis and advanced information reconstruction and error correction." Nature communications 11.1 (2020): 5345.
> > >
> > > [2] Organick, Lee, et al. "Random access in large-scale DNA data storage." Nature biotechnology 36.3 (2018): 242-248.
> > >
> > > [3] Cover, Thomas M. Elements of information theory. John Wiley & Sons, 1999.
> > >
> > > [4] Gimpel, Andreas L., et al. "A digital twin for DNA data storage based on comprehensive quantification of errors and biases." Nature Communications 14.1 (2023): 6026.

---

### Review · Reviewer_MAhQ · 2026-04-04

**Summary Of Contributions:**

Summary:
This paper formulates the trace reconstruction problem as a next-token prediction task and proposes a decoder-only transformer (TReconLM) to recover sequences from noisy traces. The method achieves state-of-the-art performance on both synthetic and real DNA storage datasets, significantly outperforming classical and neural baselines.

Strength：
1. Problem reformulation is clean and effective. Reformulating trace reconstruction as autoregressive language modeling is elegant and removes the need for explicit alignment or handcrafted priors.
2. Strong empirical performance across regimes. The method consistently outperforms classical algorithms (e.g., ITR) and neural baselines across noise levels, cluster sizes, and real datasets.
3. The paper provides both scaling law analysis and theoretical justification (e.g., connection to majority voting under substitution errors), strengthening the understanding of why transformers work.

**Audience:**

Yes

**Audience Explanation:**

Yes. The paper addresses a well-defined algorithmic problem (trace reconstruction) that is of interest to both the theoretical machine learning and applied ML communities, particularly in areas such as biological sequence analysis and DNA data storage. Beyond the specific application, the work contributes to a broader line of research on using language models for solving structured algorithmic problems.

The combination of strong empirical performance, theoretical analysis, and insights into transformer behavior makes the paper relevant to researchers studying sequence modeling, algorithmic reasoning in neural networks, and learning-based approaches to classical problems.

**Broader Impact Concerns:**

The paper does not raise immediate ethical or societal concerns. Its primary application domain is DNA data storage, which is generally considered a beneficial technology for long-term data preservation.

However, as with many advances in sequence modeling and biological data processing, there could be indirect dual-use considerations if similar techniques are applied to sensitive biological data or genomic analysis. These risks are not specific to this work but are part of a broader context in computational biology and machine learning. Overall, no significant negative broader impacts are identified.

**Claims And Evidence:**

Yes

**Claims Explanation:**

The main claims of the paper are generally well supported by clear and convincing empirical evidence. The authors conduct extensive experiments on both synthetic and real-world DNA storage datasets, comparing against a diverse set of strong baselines, including classical dynamic programming methods and recent neural approaches. The improvements are consistent across different noise levels and cluster sizes, and are evaluated using appropriate metrics such as Levenshtein distance and failure rate.

In addition, the paper provides supplementary analyses, including robustness to noise and misclustering, as well as scaling law studies and theoretical insights, which strengthen the credibility of the findings. However, some claims regarding generalization and broader applicability are less fully substantiated, as the model shows limited performance on out-of-distribution sequence lengths and relies heavily on synthetic pretraining assumptions.

**Requested Changes:**

Weakness:
1. Novelty is limited. The proposed model is essentially a standard decoder-only transformer without architectural innovation. The contribution is largely an application of existing language modeling techniques to a new problem setting. However, the analysis remains limited. The theoretical results rely on simplified assumptions and do not fully capture the challenges of real trace reconstruction (e.g., insertions and deletions). Moreover, the paper lacks deeper investigation into the model’s internal mechanisms (e.g., alignment behavior or attention patterns). As a result, the explanation is more suggestive than fully conclusive.
2. The pretraining heavily depends on a simplified synthetic error model, which may not capture complex real-world error distributions.
3. The method shows poor generalization to out-of-distribution sequence lengths (e.g., early termination issues).
4. Training requires extremely large-scale data (hundreds of billions of tokens) and compute (up to 10^20 FLOPs), which may limit practical applicability.
5. Although some comparisons with LLMs are mentioned, the evaluation lacks a thorough study of modern large multimodal or reasoning models.

Questions:
1. How sensitive is TReconLM to mismatch between synthetic pretraining error distributions and real-world error patterns?
2. Is such large-scale pretraining (hundreds of billions of tokens) necessary, or can smaller-scale training achieve comparable results?
3. How does the method compare to hybrid approaches that explicitly incorporate alignment (e.g., DP + neural models)?
4. Beyond repeated nucleotides, what are the dominant failure cases, and can they be mitigated with architectural changes?
5. Given the limited generalization to longer sequences, can the model truly be considered as learning an algorithm rather than memorizing patterns?

---

> ### Author Response · Authors · 2026-04-20
>
> We thank the reviewer for recognizing the effectiveness of our problem formulation as well as that our method provides clear improvements over the state-of-the-art. We address each of the weaknesses and questions raised below.
>
> Weaknesses:
>
> 1. **Novelty:** Regarding novelty, we agree that TReconLM does not introduce architectural innovations to the transformer. However, prior to our work it was entirely unclear whether a standard transformer trained on next-token prediction could be effectively applied to trace reconstruction. Prior deep learning methods such as RobuSeqNet and DNAformer take different approaches and do not surpass the best classical algorithms, whereas TReconLM achieves state-of-the-art performance across a range of noise levels and cluster sizes.
>
>    Regarding theoretical analysis, our analysis is indeed limited to substitution errors. The theory is limited to the substitution-only setting because this is the only setting where the Bayes-optimal estimator is known. In order to extend our current analysis approach to insertions and deletions we would first need to understand what an optimal algorithm for deletions and insertions looks like, which itself is an open problem in the field. Regarding internal mechanisms, we visualize attention maps across all layers in Appendix D, finding that early layers perform coarse alignment while later layers show more focused diagonals, suggesting that the model progressively refines its alignment.
>
> 2. **Synthetic pretraining:** We choose a uniform random error model for pretraining because it is technology-agnostic and can later be finetuned comparably cheaply to the error patterns of a specific technology. The uniform assumption also holds approximately in practice [1], which can be seen from TReconLM outperforming DNAformer (failure rate 0.111 vs. 0.146) in Appendix J.2, even though DNAformer was pretrained on synthetic data generated from the actual Microsoft dataset error statistics, suggesting that a uniform error model is also an effective pretraining strategy.
>
> 3. **Poor generalization to out-of-distribution sequence lengths:**  We acknowledge that the model shows limited generalization to out-of-distribution sequence lengths, including early termination for longer sequences. However, all current DNA storage systems fall within a relatively narrow range (50-200 nt) due to current synthesis constraints (e.g. [2] 190nt, [3] 194nt, [4] 117nt, [5] 152nt, [6] 150nt, [7] 159nt, [8] 120nt, [9] 158nt, [10] 102nt, [11] 140nt ). It is considered unlikely that practical systems will exceed this range in the foreseeable future, especially for cost-efficient systems [1]. In Section 5.2.4, we intentionally trained only on $L \in [50, 120]$ to test out-of-distribution generalization to longer sequences. Training on the full practical range $L \in [50, 200]$ would resolve the early termination issue at $L=180$.
>
> 4. **Practical applicability due to training cost:** While we use $10^{20}$ FLOPs for full pretraining, our scaling law analysis (Section 5.4) shows that strong performance can already be achieved with relatively little compute. A model trained with $3\times10^{18}$ FLOPs (approximately 19 A100 GPU hours) already achieves a failure rate of $64.09\%$ and Levenshtein distance of $0.0399$, and a model trained with $10^{19}$ FLOPs (approximately 65 GPU hours) achieves $57.45\%$ and $0.0356$, compared to $53.66\%$ and $0.0335$ for the full $10^{20}$ FLOPs model (approximately 200 A100 GPU hours). Moreover, pretraining is a one-time cost, and the pretrained model can serve as a foundation model requiring relatively little fine-tuning data to adapt to specific DNA data storage technologies. As shown in Appendix H, using only $5\%$ of the fine-tuning data already reduces the failure rate from $0.999$ to $0.887$ on the Noisy-DNA dataset. The small optimal model size (approximately 38M parameters) also means that inference is cheap. TReconLM achieves comparable throughput per hour and dollar to MUSCLE, which is widely used in practice for sequence alignment (Appendix L). We therefore do not expect compute requirements to limit the practical applicability of TReconLM.
>
> 5. **Evaluation against large multimodal or reasoning models:** In Appendix J.3 we compare TReconLM against GPT-4o mini and GPT-5, evaluating both models under zero-, three-, and five-shot prompting with and without Chain-of-Thought reasoning. We find that TReconLM outperforms both models across all cluster sizes, and that reasoning does not improve reconstruction accuracy for either model. Regarding multimodal models, trace reconstruction is a purely text-based task over a four-character nucleotide alphabet, and we therefore do not see a clear motivation for evaluating multimodal models.

---

> ### Author Response · Authors · 2026-04-20
>
> Questions:
>
> 1. **Sensitivity to distribution mismatch and generalization:** The sensitivity depends on the degree of mismatch. For the Microsoft dataset, the pretrained model generalizes well without fine-tuning, achieving performance comparable to fine-tuned DNAformer and outperforming all non-deep learning baselines. For the Noisy-DNA dataset, which has position-dependent error patterns (insertion probability increasing up to $p_I = 0.3$ toward sequence ends), we observe larger gains from fine-tuning. To further assess generalization, we additionally ran cross-dataset experiments in which models fine-tuned on one dataset are evaluated on another, but did not observe cross-dataset generalization (see also our response to Question 2 from Reviewer EQ1k for a detailed overview table of results).
>
> 2. **Dataset size for pretraining:** Our scaling law analysis (Section 5.4) addresses this tradeoff. As with language models, test loss follows a power law relationship with the number of training tokens, meaning performance improves with more training tokens. Large-scale pretraining is therefore not strictly necessary, but rather a performance choice.
>
>    To make this tradeoff more concrete, we additionally report the Levenshtein distance and failure rate at varying training token counts (always selecting the model with the lowest loss per compute budget), averaged over cluster sizes $N \in [2, 10]$:
>
>    | Metric               | 1.75M | 3.14M | 3.84M | 8.81M | 17.66M | 29.37M | 88.10M | 176.61M |
>    |----------------------|-------|-------|-------|-------|--------|--------|--------|---------|
>    | Levenshtein distance | 0.060 | 0.051 | 0.044 | 0.041 | 0.037  | 0.035  | 0.034  | 0.033   |
>    | Failure rate         | 0.801 | 0.747 | 0.756 | 0.751 | 0.598  | 0.571  | 0.546  | 0.534   |
>
> 3. **Comparison to hybrid approaches:** In Appendix J.2, we compare against DNAformer's self-reported performance on the Microsoft dataset. DNAformer's reported performance includes their CPL algorithm, a DP-based fallback that is used when the neural model has low confidence, and is therefore a hybrid approach that combines neural methods with DP. We find that TReconLM outperforms DNAformer despite using fewer reads. We are not aware of other hybrid DP+neural approaches for trace reconstruction, but would be happy to include additional comparisons if the reviewer has a specific method in mind. We additionally tested whether explicit alignment might give the transformer cleaner input to work with by training TReconLM on multiple sequence alignments instead of raw traces (Appendix A), but find that this performs worse than direct reconstruction.
>
> 4.  **Failure modes and architectural mitigation.** Beyond runs of consecutive identical nucleotides (enrichment 1.41), near-repetitive patterns of the form XYY, YXYYY, and ZYXYYYY show higher error rates than expected (Table 2, Appendix D). We do not believe architectural changes are the most effective mitigation, and data augmentation techniques such as oversampling near-repetitive patterns are more promising, for the following reason.
> Theoretically, alternating strings (XYX, XYZ) are the hardest sequences to reconstruct, while repeated nucleotides are the easiest [12]. We also verified this empirically via exact posteriors for short sequences ($L=12$, $N=2$, 1K samples), finding enrichment factors of $1.25\times$ (XYX) and $1.23\times$ (XYZ) for alternating patterns, while near-repetitive patterns such as XZZ have a lower enrichment factor ($0.91\times$), and repeated nucleotides are easiest ($0.28\times$). On the other hand, TReconLM reconstructs alternating patterns at near-uniform difficulty (XYX: $1.03\times$, XYZ: $1.01\times$), while near-repetitive patterns are harder (e.g., XYY has enrichment $1.51\times$). We explain this by the fact that under uniform data generation, alternating strings are the most probable patterns to be generated, so the model sees many examples during training and learns to reconstruct them well. Near-repetitive patterns occur less frequently under uniform data generation and are harder to reconstruct than repeated nucleotides, so both their lower training frequency and inherent reconstruction difficulty contribute to them being the most challenging patterns for the model. Therefore, we believe that a data generation pipeline balancing pattern frequency against reconstruction difficulty would lead to more uniform reconstruction error rates, though we expect this to matter less at large training scales.

---

> > ### Author Response · Authors · 2026-04-20
> >
> > 5. **Learning an algorithm vs. memorizing patterns:** We believe that poor generalization to out-of-distribution sequence lengths does not necessarily contradict that TReconLM can learn a meaningful reconstruction algorithm rather than just memorizing patterns. Existing classical algorithms such as TrellisBMA also explicitly use the ground truth sequence length in their algorithm, and the ground truth sequence length is typically known in trace reconstruction.
> >
> > We hope this addresses the reviewer’s concern and are happy to clarify any remaining points further.
> >
> > [1] Gimpel, Andreas L., et al. "A digital twin for DNA data storage based on comprehensive quantification of errors and biases." Nature Communications (2023).
> >
> > [2] Wang, Yixin, et al. "High capacity DNA data storage with variable-length Oligonucleotides using repeat accumulate code and hybrid mapping." Journal of biological engineering (2019).
> >
> > [3] Anavy, Leon, et al. "Data storage in DNA with fewer synthesis cycles using composite DNA letters." Nature biotechnology (2019).
> >
> > [4]Goldman, Nick, et al. "Towards practical, high-capacity, low-maintenance information storage in synthesized DNA." Nature (2013).
> >
> > [5] Erlich, Yaniv, and Dina Zielinski. "DNA Fountain enables a robust and efficient storage architecture." Science (2017).
> >
> > [6] Organick, Lee, et al. "Random access in large-scale DNA data storage." Nature biotechnology (2018).
> >
> > [7] Church, George M., Yuan Gao, and Sriram Kosuri. "Next-generation digital information storage in DNA." Science (2012).
> >
> > [8] Bornholt, James, et al. "A DNA-based archival storage system." Proceedings of the twenty-first international conference on architectural support for programming languages and operating systems. (2016).
> >
> > [9] Grass, Robert N., et al. "Robust chemical preservation of digital information on DNA in silica with error‐correcting codes." Angewandte Chemie International Edition (2015).
> >
> > [10] Meiser, Linda C., et al. "Synthetic DNA applications in information technology." Nature communications (2022).
> >
> > [11] Bar-Lev, Daniella, et al. "Scalable and robust DNA-based storage via coding theory and deep learning." Nature Machine Intelligence (2025).
> >
> > [12] Holden, Nina, and Russell Lyons. "Lower bounds for trace reconstruction." (2020): 503-525.

---

### Review · Reviewer_yPLZ · 2026-04-12

**Summary Of Contributions:**

This paper studies the noisy code reconstruction problem in a DNA data storage setting.
TReconLM is proposed a decoder only transformer, which is trained to reconstruct DNA-sequences.
In the experimental section a transformer is initially pre-trained using synthetic sequences.
Later, a fine-tuning step is run on real data.

The paper claims state of the art performance.
After observing state-of the art performance the experimental section moves on to studying scaling laws for trace reconstruction. An optimum is found at approximately 38 Million parameters.

Finally the paper examines the scaling behavior theoretically considering substitution errors.

**Audience:**

Yes

**Audience Explanation:**

Yes, I think this work is relevant to the bioinformatics community.

**Claims And Evidence:**

Yes

**Claims Explanation:**

Yes the experimental section backs up the papers claim of state of the art performance a comparison to seven related methods.

The paper's claim of an optimal model size is backed by a large scale study of running various model sizes.

- I failed to understand figure two. Where are the shaded bands and error bars?

**Requested Changes:**

- Code, Thanks for shipping code with this paper. Its nice that a conda compatible yml file specifies the codes' dependencies.
Systematic tests ( https://docs.pytest.org/en/stable/ nox) could strengthen the project's credibility further, potentially even containerized using i.e. nox (https://nox.thea.codes/en/stable/).
- Improve figure two. Where are the error bars?
- Is it possible to elaborate on the parameter counts of the methods the paper compares in the related section? Are all methods using similar numbers of trainable parameters? Are compute needs in the same ballpark?
- A Table comparing inference times for the various models could strengthen the paper further.

---

> ### Author Response · Authors · 2026-04-20
>
> We thank the reviewer for their positive assessment of our work and address each requested change below.
>
> 1. **Code:** We agree that systematic tests improve the reproducibility and usability of the code. We have added a test suite using pytest (in tests/) for our data generation pipeline (error model, alignment and string nesting), tokenization (single sequence encoding/decoding, batched encoding/decoding and padding), for model training (expected logit shape, assertion when input length exceeds block size, short training run to see if loss decreases) and inference (KV cache and masking should not change output, majority voting, permutation of input traces and beam search works as expected). We have also added a noxfile.py so the test suite can be run with a single command.
>
>    Additionally, we have uploaded our pretrained and fine-tuned model weights to an anonymized HuggingFace repository, and added new tutorial notebooks (tutorial/quick_start.ipynb and tutorial/custom_data.ipynb) to download the weights and run inference on the synthetic and real world test data. We hope this makes it easier to reproduce and build on our results.
>
> 2. **Figure 2:** We agree that the shaded bands are difficult to see in Figure 2. They are only visible when zoomed in, as TReconLM has very low variance across the three runs with different seeds (we have added this clarification to the figure caption). For example, the mean failure rate across all cluster sizes is $53.66\%$, $53.50\%$, and $53.43\%$ across the three seeds and the Levenshtein distance is $0.0335$, $0.0334$, and $0.0334$. To improve visibility, we have also made the error bars black in all failure rate plots.
>
> 3. **Parameter count of baselines:** The baseline methods vary in parameter count. RobuSeqNet has approximately 3M parameters, TReconLM approximately 38M, and DNAformer approximately 100M. We therefore match the number of training examples rather than compute, as pretraining the small 3M-parameter baseline for reconstructing sequences of length $L=110$ for matched compute would be impractical. To see this, we measured training throughput using a 1K-iteration run (skipping the first 100 warmup iterations) with PyTorch's profiler to estimate per-iteration FLOPs. After warmup, the model processes about $1.125 \times 10^{13}$ FLOPs per second on a single A100, corresponding to roughly 103 days to reach a $10^{20}$-FLOP budget, or approximately 13 days on 8 GPUs assuming no communication overhead. To control for parameter count and compute, we additionally compare TReconLM with RobuSeqNet under smaller, matched compute ($6 \times 10^{17}$ FLOPs) and model size (approximately 3M parameters) in Appendix J.1, and find that TReconLM still outperforms RobuSeqNet.
>
>    For DNAformer, we estimated the total pretraining compute for reconstructing sequences of length $L=110$ using the same profiling approach and get approximately $5 \times 10^{19}$ FLOPs, compared to $1.0 \times 10^{20}$ FLOPs for TReconLM. As an additional matched-compute comparison, we evaluated TReconLM at a smaller compute budget of $1 \times 10^{19}$ FLOPs using the best model size (approximately 38M parameters) from our scaling law experiments and get a failure rate of $57.45\%$ and Levenshtein distance of $0.0356$, compared to $53.66\%$ and $0.0335$ for TReconLM at $1 \times 10^{20}$ FLOPs. DNAformer achieves a failure rate of $61.51\%$ and Levenshtein distance of $0.0682$, showing that TReconLM also outperforms DNAformer at a smaller compute budget and with fewer training examples. In Appendix J.2 we additionally compare to the self-reported performance of DNAformer on the Microsoft dataset, where TReconLM also performs better.

---

> > ### Author Response · Authors · 2026-04-20
> >
> > 4. **Inference time:** We agree that inference time is an important practical consideration. We discuss inference cost in Appendix L and Figure 16, which reports throughput per hour and per dollar for all methods. Throughput per dollar is computed by dividing throughput per hour by the hardware cost to more fairly compare between CPU and GPU based methods (USD 3.40/hr for one A100 GPU and USD 4.00/hr for 64 CPUs, based on Azure pricing). We additionally include a summary table here for convenience:
> >
> >    | Method | Throughput (ex/hr) | Throughput (ex/\$) |
> >    |--------|-------------------:|-------------------:|
> >    | VS | 19,027,811 | 4,756,953 |
> >    | BMALA | 2,273,193 | 568,298 |
> >    | DNAformer | 874,953 | 257,339 |
> >    | RobuSeqNet | 719,927 | 211,743 |
> >    | ITR | 644,882 | 161,221 |
> >    | TReconLM | 291,601 | 85,765 |
> >    | MUSCLE | 264,051 | 66,013 |
> >    | TrellisBMA | 2,990 | 748 |
> >
> >    TReconLM achieves similar throughput to MUSCLE, which is widely used in practice for sequence alignment. It is nearly two orders of magnitude faster than TrellisBMA, which relies on a much more expensive BCJR-based trellis computation. TReconLM is slower than DNAformer and RobuSeqNet, which predict the entire sequence in a single forward pass, but achieves better reconstruction accuracy. If inference speed becomes critical, transformer acceleration techniques such as speculative decoding could further improve TReconLM's throughput.
> >
> > Please let us know if this addressed your concerns; we’re happy to clarify further.

---

> > > ### Comment · Reviewer_yPLZ · 2026-04-21
> > > **Thank you**
> > >
> > > Thank you for responding and for cleaning up the code. I have no further questions.

---

### Decision · Action_Editor_ADJP · 2026-06-03

**Recommendation:** Accept as is

**Additional Comments:**

This is a nice paper that tackles a problem on DNA reconstruction that has been traditionally studied in bioinformatics and information theory. The idea is that sequences from some alphabet (of size 4 for the DNA case) are affected by insertions, deletions, and substitutions, and we observe some number of corrupted samples (traces). The authors train a model to reconstruct the original sequence from these traces. This is done rather than use one of the existing hand-crafted algorithms.

The authors’ decoder-only Transformer model performs very well, beating out these traditional baselines; the authors also introduce a nice analysis. All of the reviewers are broadly positive. I agree with them; the paper is a nice contribution that shows learned approaches can beat hand-crafted strategies.

**Audience:**

Yes

**Audience Explanation:**

Yes, the paper uses language modeling techniques of interest to the TMLR audience; it also tackles bioinformatics problems of interest to the community.

**Claims And Evidence:**

Yes

**Claims Explanation:**

The paper has experimental and theoretical support for its claims.